# PAC Confidence Predictions for Deep Neural Network Classifiers

**Sangdon Park, Shuo Li, Insup Lee & Osbert Bastani**
PRECISE Center
University of Pennsylvania
`{sangdonp, lishuo1, lee, obastani}@seas.upenn.edu`

### Abstract

A key challenge for deploying deep neural networks (DNNs) in safety critical settings is the need to provide rigorous ways to quantify their uncertainty. In this paper, we propose a novel algorithm for constructing predicted classification confidences for DNNs that comes with provable correctness guarantees. Our approach uses Clopper-Pearson confidence intervals for the Binomial distribution in conjunction with the histogram binning approach to calibrated prediction. In addition, we demonstrate how our predicted confidences can be used to enable downstream guarantees in two settings: (i) fast DNN inference, where we demonstrate how to compose a fast but inaccurate DNN with an accurate but slow DNN in a rigorous way to improve performance without sacrificing accuracy, and (ii) safe planning, where we guarantee safety when using a DNN to predict whether a given action is safe based on visual observations. In our experiments, we demonstrate that our approach can be used to provide guarantees for state-of-the-art DNNs.

## 1 Introduction

Due to the recent success of machine learning, there has been increasing interest in using predictive models such as deep neural networks (DNNs) in safety-critical settings, such as robotics (e.g., obstacle detection (Ren et al., 2015) and forecasting (Kitani et al., 2012)) and healthcare (e.g., diagnosis (Gulshan et al., 2016; Esteva et al., 2017) and patient care management (Liao et al., 2020)).

One of the key challenges is the need to provide guarantees on the safety or performance of DNNs used in these settings. The potential for failure is inevitable when using DNNs, since they will inevitably make some mistakes in their predictions. Instead, our goal is to design tools for *quantifying* the uncertainty of these models; then, the overall system can estimate and account for the risk inherent in using the predictions made by these models. For instance, a medical decision-making system may want to fall back on a doctor when its prediction is uncertain whether its diagnosis is correct, or a robot may want to stop moving and ask a human for help if it is uncertain to act safely. Uncertainty estimates can also be useful for human decision-makers—*e.g.,* for a doctor to decide whether to trust their intuition over the predicted diagnosis.

While many DNNs provide confidences in their predictions, especially in the classification setting, these are often overconfident. This phenomenon is most likely because DNNs are designed to overfit the training data (*e.g.,* to avoid local minima (Safran & Shamir, 2018)), which results in the predicted probabilities on the training data being very close to one for the correct prediction. Recent work has demonstrated how to *calibrate* the confidences to significantly reduce overconfidence (Guo et al., 2017). Intuitively, these techniques rescale the confidences on a held-out calibration set. Because they are only fitting a small number of parameters, they do not overfit the data as was the case in the original DNN training. However, these techniques do not provide theoretical guarantees on their correctness, which can be necessary in safety-critical settings to guarantee correctness.

We propose a novel algorithm for calibrated prediction in the classification setting that provides theoretical guarantees on the predicted confidences. We focus on *on-distribution* guarantees—*i.e.,* where the test distribution is the same as the training distribution. In this setting, we can build on ideas from statistical learning theory to provide *probably approximately correctness (PAC)* guarantees (Valiant, 1984). Our approach is based on a calibrated prediction technique called histogram

binning (Zadrozny & Elkan, 2001), which rescales the confidences by binning them and then rescaling each bin independently. We use Clopper-Pearson bounds on the tails of the binomial distribution to obtain PAC upper/lower bounds on the predicted confidences.

Next, we study how it enables theoretical guarantees in two applications. First, we consider the problem of speeding up DNN inference by composing a fast but inaccurate model with a slow but accurate model—*i.e.,* by using the accurate model for inference only if the confidence of the inaccurate one is underconfident (Teerapittayanon et al., 2016). We use our algorithm to obtain guarantees on accuracy of the composed model. Second, for safe planning, we consider using a DNN to predict whether or not a given action (*e.g.,* move forward) is safe (*e.g.,* do not run into obstacles) given an observation (*e.g.,* a camera image). The robot only continues to act if the predicted confidence is above some threshold. We use our algorithm to ensure safety with high probability. Finally, we evaluate the efficacy of our approach in the context of these applications.

**Related work.** Calibrated prediction (Murphy, 1972; DeGroot & Fienberg, 1983; Platt, 1999) has recently gained attention as a way to improve DNN confidences (Guo et al., 2017). Histogram binning is a non-parametric approach that sorts the data into finitely many bins and rescales the confidences per bin (Zadrozny & Elkan, 2001; 2002; Naeini et al., 2015). However, traditional approaches do not provide theoretical guarantees on the predicted confidences. There has been work on predicting confidence sets (*i.e.,* predict a set of labels instead of a single label) with theoretical guarantees (Park et al., 2020a), but this approach does not provide the confidence of the most likely prediction, as is often desired. There has also been work providing guarantees on the overall calibration error (Kumar et al., 2019), but this approach does not provide per-prediction guarantees.

There has been work speeding up DNN inference (Hinton et al., 2015). One approach is to allow intermediate layers to be dynamically skipped (Teerapittayanon et al., 2016; Figurnov et al., 2017; Wang et al., 2018), which can be thought of as composing multiple models that share a backbone. Unlike our approach, they do not provide guarantees on the accuracy of the composed model.

There has also been work on safe learning-based control (Akametalu et al., 2014; Fisac et al., 2019; Bastani, 2019; Li & Bastani, 2020; Wabersich & Zeilinger, 2018; Alshiekh et al., 2018); however, these approaches are not applicable to perception-based control. The most closely related work is Dean et al. (2019), which handles perception, but they are restricted to known linear dynamics.

## 2 PAC CONFIDENCE PREDICTION

In this section, we begin by formalizing the PAC confidence coverage prediction problem; then, we describe our algorithm for solving this problem based on histogram binning.

**Calibrated prediction.** Let $x \in \mathcal{X}$ be an example and $y \in \mathcal{Y}$ be one of a finite label set, and let $D$ be a distribution over $\mathcal{X} \times \mathcal{Y}$. A *confidence predictor* is a model $\hat{f} : \mathcal{X} \to \mathcal{P}_{\mathcal{Y}}$, where $\mathcal{P}_{\mathcal{Y}}$ is the space of probability distributions over labels. In particular, $\hat{f}(x)_y$ is the predicted confidence that the true label for $x$ is $y$. We let $\hat{y} : \mathcal{X} \to \mathcal{Y}$ be the corresponding *label predictor*—*i.e.,* $\hat{y}(x) := \arg\max_{y \in \mathcal{Y}} \hat{f}(x)_y$—and let $\hat{p} : \mathcal{X} \to \mathbb{R}_{\geq 0}$ be corresponding *top-label confidence predictor*—*i.e.,* $\hat{p}(x) := \max_{y \in \mathcal{Y}} \hat{f}(x)_y$. While traditional DNN classifiers are confidence predictors, a naively trained DNN is not reliable—*i.e.,* predicted confidence does not match to the true confidence; recent work has studied heuristics for improving reliability (Guo et al., 2017). In contrast, our goal is to construct a confidence predictor that comes with theoretical guarantees.

We first introduce the definition of calibration (DeGroot & Fienberg, 1983; Zadrozny & Elkan, 2002; Park et al., 2020b)—*i.e.,* what we mean for a predicted confidence to be "correct". In many cases, the main quantity of interest is the confidence of the top prediction. Thus, we focus on ensuring that the top-label predicted confidence $\hat{p}(x)$ is calibrated (Guo et al., 2017); our approach can easily be extended to providing guarantees on all confidences predicted using $\hat{f}$. Then, we say a confidence predictor $\hat{f}$ is *well-calibrated* with respect to distribution $D$ if

$$\mathbb{P}_{(x,y)\sim D}\left[y = \hat{y}(x) \mid \hat{p}(x) = t\right] = t \qquad (\forall t \in [0, 1]).$$

That is, among all examples $x$ such that the label prediction $\hat{y}(x)$ has predicted confidence $t = \hat{p}(x)$, $\hat{y}(x)$ is the correct label for exactly a $t$ fraction of these examples. Using a change of variables (Park

et al., 2020b), $\hat{f}$ being well-calibrated is equivalent to

$$\hat{p}(x) = c_{\hat{f}}^*(x) := \mathbb{P}_{(x',y')\sim D}\left[y' = \hat{y}(x') \mid \hat{p}(x') = \hat{p}(x)\right] \qquad (\forall x \in \mathcal{X}). \qquad (1)$$

Then, the goal of well-calibration is to make $\hat{p}$ equal to $c_{\hat{f}}^*$. Note that $\hat{f}$ appears on both sides of the equation $\hat{p}(x) = c_{\hat{f}}^*(x)$—implicitly in $\hat{p}$—which is what makes it challenging to satisfy. Indeed, in general, it is unlikely that (1) holds exactly for all $x$. Instead, based on the idea of histogram binning (Zadrozny & Elkan, 2001), we consider a variant where we partition the data into a fixed number of bins and then construct confidence coverages separately for each bin. In particular, consider $K$ bins $B_1, \ldots, B_K \subseteq [0,1]$, where $B_1 = [0, b_1]$ and $B_k = (b_{k-1}, b_k]$ for $k > 1$. Here, $K$ and $0 \le b_1 \le \cdots \le b_{K-1} \le b_K = 1$ are hyperparameters. Given $\hat{f}$, let $\kappa_{\hat{f}} : \mathcal{X} \to \{1, \ldots, K\}$ to denote the index of the bin that contains $\hat{p}(x)$—i.e., $\hat{p}(x) \in B_{\kappa_{\hat{f}}(x)}$.

**Definition 1** We say $\hat{f}$ is *well-calibrated* for a distribution $D$ and bins $B_1, \ldots, B_K$ if

$$\hat{p}(x) = c_{\hat{f}}(x) := \mathbb{P}_{(x',y')\sim D}\left[y' = \hat{y}(x') \,\Big|\, \hat{p}(x') \in B_{\kappa_{\hat{f}}(x)}\right] \qquad (\forall x \in \mathcal{X}), \qquad (2)$$

where we refer to $c_{\hat{f}}(x)$ as the *true confidence*. Intuitively, this definition "coarsens" the calibration problem across the bins—i.e., rather than sorting the inputs $x$ into a continuum of "bins" $\hat{p}(x) = t$ for each $t \in [0,1]$ as in (1), we sort them into a finite number of bins $\hat{p}(x) \in B_k$; intuitively, we have $c_{\hat{f}}^* \approx c_{\hat{f}}$ if the bin sizes are close to zero. It may not be obvious what downstream guarantees can be obtained based on this definition; we provide examples in Sections 3 & 4.

**Problem formulation.** We formalize the problem of PAC calibration. We focus on the setting where the training and test distributions are identical—e.g., we cannot handle adversarial examples or changes in covariate distribution (e.g., common in reinforcement learning). Importantly, while we assume a pre-trained confidence predictor $\hat{f}$ is given, we make no assumptions about $\hat{f}$—e.g., it can be uncalibrated or heuristically calibrated. If $\hat{f}$ performs poorly, then the predicted confidences will be close to $1/|\mathcal{Y}|$—i.e., express no confidence in the predictions. Thus, it is fine if $\hat{f}$ is poorly calibrated; the important property is that the confidence predictor $\hat{f}$ have similar true confidences.

The challenge in formalizing PAC calibration is that quantifying over all $x$ in (2). One approach is to provide guarantees in expectation over $x$ (Kumar et al., 2019); however, this approach does not provide guarantees for individual predictions.

Instead, our goal is to find a *set* of predicted confidences that includes the true confidence with high probability. Of course, we could simply predict the interval $[0,1]$, which always contains the true confidence; thus, simultaneously want to make the size of the interval small. To this end, we consider a *confidence coverage predictor* $\hat{C} : \mathcal{X} \to 2^{\mathbb{R}}$, where $c_{\hat{f}}(x) \in \hat{C}(x)$ with high probability. In particular, $\hat{C}(x)$ outputs an interval $[\underline{c}, \overline{c}] \subseteq \mathbb{R}$, where $\underline{c} \le \overline{c}$, instead of a set. We only consider a single interval (rather than disjoint intervals) since one suffices to localize the true confidence $c_{\hat{f}}$.

We are interested in providing theoretical guarantees for an algorithm used to construct confidence coverage predictor $\hat{C}$ given a held-out calibration set $Z \subseteq \mathcal{X} \times \mathcal{Y}$. In addition, we assume the algorithm is given a pretrained confidence predictor $\hat{f}$. Thus, we consider $\hat{C}$ as depending on $Z$ and $\hat{f}$, which we denote by $\hat{C}(\cdot; \hat{f}, Z)$. Then, we want $\hat{C}$ to satisfy the following guarantee:

**Definition 2** Given $\delta \in \mathbb{R}_{>0}$ and $n \in \mathbb{N}$, $\hat{C}$ is *probably approximately correct (PAC)* if for any $D$,

$$\mathbb{P}_{Z\sim D^n}\left[\bigwedge_{x\in\mathcal{X}} c_{\hat{f}}(x) \in \hat{C}(x; \hat{f}, Z)\right] \ge 1 - \delta. \qquad (3)$$

Note that $c_{\hat{f}}$ depends on $D$. Here, "approximately correct" technically refers to the mean of $\hat{C}(x; \hat{f}, Z)$, which is an estimate of $c_{\hat{f}}(x)$; the interval captures the bound $\epsilon$ on the error of this estimate; see Appendix A for details. Furthermore, the conjunction over all $x \in \mathcal{X}$ may seem strong. We can obtain such a guarantee due to our binning strategy: the property $c_{\hat{f}}(x) \in \hat{C}(x; \hat{f}, Z)$ only depends on the bin $B_{\kappa_{\hat{f}}(x)}$, so the conjunction is really only over bins $k \in \{1, ..., K\}$.

**Algorithm.** We propose a confidence coverage predictor that satisfies the PAC property. The problem of estimating the confidence interval $\hat{C}(x)$ of the binned true confidence $c_{\hat{f}}(x)$ is closely related to the binomial proportion confidence interval estimation; consider a Bernoulli random variable $b \sim B := \text{Bernoulli}(\theta)$ for any $\theta \in [0, 1]$, where $b = 1$ denotes a success and $b = 0$ denotes a failure, and $\theta$ is unknown. Given a sequence of observations $b_{1:n} := (b_1, \ldots, b_n) \sim B^n$, the goal is to construct an interval $\hat{\Theta}(b_{1:n}) \subseteq \mathbb{R}$ that includes $\theta$ with high probability—i.e.,

$$\mathbb{P}_{b_{1:n} \sim B^n} \left[ \theta \in \hat{\Theta}(b_{1:n}) \right] \geq 1 - \alpha, \tag{4}$$

where $\alpha \in \mathbb{R}_{>0}$ is a given confidence level. In particular, the Clopper-Pearson interval

$$\hat{\Theta}_{\text{CP}}(b_{1:n}; \alpha) := \left[ \inf_{\theta} \left\{ \theta \,\Big|\, \mathbb{P}_\theta \left[ S \geq s \right] \geq \frac{\alpha}{2} \right\}, \; \sup_{\theta} \left\{ \theta \,\Big|\, \mathbb{P}_\theta \left[ S \leq s \right] \geq \frac{\alpha}{2} \right\} \right],$$

guarantees (4) (Clopper & Pearson, 1934; Brown et al., 2001), where $s = \sum_{i=1}^{n} b_i$ is the number of observed successes, $n$ is the number of observations, and $S$ is a Binomial random variable $S \sim \text{Binomial}(n, \theta)$. Intuitively, the interval is constructed such that the number of observed success falls in the region with high-probability for any $\theta$ in the interval. The following expression is equivalent due to the relationship between the Binomial and Beta distributions (Hartley & Fitch, 1951; Brown et al., 2001)—i.e., $\mathbb{P}_\theta[S \geq s] = I_\theta(s, n - s + 1)$, where $I_\theta$ is the CDF of $\text{Beta}(s, n - s + 1)$:

$$\hat{\Theta}_{\text{CP}}(b_{1:n}; \alpha) = \left[ \frac{\alpha}{2} \text{ quantile of Beta}(s, n - s + 1), \; \left(1 - \frac{\alpha}{2}\right) \text{ quantile of Beta}(s + 1, n - s) \right].$$

Now, for each of the $K$ bins, we apply $\hat{\Theta}_{\text{CP}}$ with confidence level $\alpha = \frac{\delta}{K}$—i.e.,

$$\hat{C}(x; \hat{f}, Z, \delta) := \hat{\Theta}_{\text{CP}} \left( W_{\kappa_{\hat{f}}(x)}; \frac{\delta}{K} \right) \text{ where } W_k := \left\{ \mathbb{1}(\hat{y}(x) = y) \,\Big|\, (x, y) \in Z \text{ s.t. } \kappa_{\hat{f}}(x) = k \right\}.$$

Here, $W_k$ is the set of observations of successes vs. failures corresponding to the subset of labeled examples $(x, y) \in Z$ such that $\hat{p}(x)$ falls in the bin $B_k$, where a success is defined to be a correct prediction $\hat{y}(x) = y$. We note that for efficiency, the confidence interval for each of the $K$ bins can be precomputed. Our construction of $\hat{C}$ satisfies the following; see Appendix B for a proof:

**Theorem 1** *Our confidence coverage predictor $\hat{C}$ is PAC for any $\delta \in \mathbb{R}_{>0}$ and $n \in \mathbb{N}$.*

Note that Clopper-Pearson intervals are exact, ensuring the size of $\hat{C}$ for each bin is small in practice. Finally, an important special case is when there is a single bin $B = [0, 1]$—i.e.,

$$\hat{C}_0(x; \hat{f}, Z', \delta) := \hat{\Theta}_{\text{CP}}(W; \delta) \qquad \text{where} \qquad W := \{\mathbb{1}(\hat{y}(x') = y') \mid (x', y') \in Z'\}.$$

Note that $\hat{C}_0$ does not depend on $x$, so we drop it—i.e., $\hat{C}_0(\hat{f}, Z', \delta) := \hat{\Theta}_{\text{CP}}(W; \delta)$—i.e., $\hat{C}_0$ computes the Clopper-Pearson interval over $Z'$, which is a subset of the original set $Z$.

## 3 APPLICATION TO FAST DNN INFERENCE

A key application of predicted confidences is to perform *model composition* to improve the running time of DNNs without sacrificing accuracy. The idea is to use a fast but inaccurate model when it is confident in its prediction, and switch to an accurate but slow model otherwise (Bolukbasi et al., 2017); we refer to the combination as the *composed model*. To further improve performance, we can have the two models share a backbone—i.e., the fast model shares the first few layers of the slow model (Teerapittayanon et al., 2016). We refer to the decision of whether to skip the slow model as the *exit condition*; then, our goal is to construct confidence thresholds for exit conditions in a way that provides theoretical guarantees on the overall accuracy.

**Problem setup.** The early-stopping approach for fast DNN inference can be formalized as a sequence of branching classifiers organized in a cascading way—i.e.,

$$\hat{y}_C(x; \gamma_{1:M-1}) := \begin{cases} \hat{y}_m(x) & \text{if } \bigwedge_{i=1}^{m-1} (\hat{p}_i(x) < \gamma_i) \wedge (\hat{p}_m(x) \geq \gamma_m) \; (\forall m \in \{1, \ldots, M-1\}) \\ \hat{y}_M(x) & \text{otherwise,} \end{cases}$$

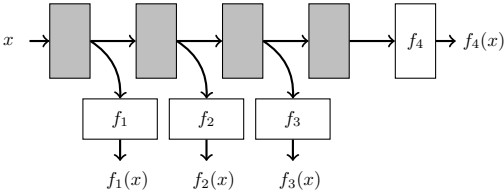

Figure 1: A composed model in a cascading way for $M = 4$.

where $M$ is the number of branches, $\hat{f}_m$ is the confidence predictor, and $\hat{y}_m$ and $\hat{p}_m$ are the associated label and top-label confidence predictor, respectively. For conciseness, we denote the exit condition of the $m$th branch by $d_m$ (i.e., $d_m(x) := \mathbb{1}(\bigwedge_{i=1}^{m-1} (\hat{p}_i(x) < \gamma_i) \wedge (\hat{p}_m(x) \geq \gamma_m)))$ with thresholds $\gamma_1, \ldots, \gamma_m \in [0, 1]$. The $\hat{f}_m$ share a backbone and are trained in the standard way; see Appendix F.4 for details. Figure 1 illustrates the composed model for $M = 4$; the gray area represents the shared backbone. We refer to an overall model composed in this way as a *cascading classifier*.

**Desired error bound.** Given $\xi \in \mathbb{R}_{>0}$, our goal is to choose $\gamma_{1:M-1} := (\gamma_1, \ldots, \gamma_{M-1})$ so the error difference of the cascading classifier $\hat{y}_C$ and the slow classifier $\hat{y}_M$ is at most $\xi$—i.e.,

$$p_{\text{err}} := \mathbb{P}_{(x,y) \sim D} [\hat{y}_C(x) \neq y] - \mathbb{P}_{(x,y) \sim D} [\hat{y}_M(x) \neq y] \leq \xi. \tag{5}$$

To obtain the desired error bound, an example $x$ exits at the $m$th branch if $\hat{y}_m$ is likely to classify $x$ correctly, allowing for at most $\xi$ fraction of errors total. Intuitively, if the confidence of $\hat{y}_m$ on $x$ is sufficiently high, then $\hat{y}_m(x) = y$ with high probability. In this case, $\hat{y}_M$ either correctly classifies or misclassifies the same example; if the example is misclassified, it contributes to decrease $p_{\text{err}}$; otherwise, we have $\hat{y}_m(x) = y = \hat{y}_M(x)$ with high probability, which contributes to maintain $p_{\text{err}}$.

**Fast inference.** To minimize running time, we prefer to allow higher error rates at the lower branches—i.e., we want to choose $\gamma_m$ as small as possible at lower branches $m$.

**Algorithm.** Our algorithm takes prediction branches $\hat{f}_m$ (for $m \in \{1, \ldots, M\}$), the desired relative error $\xi \in [0, 1]$, a confidence level $\delta \in [0, 1]$, and a calibration set $Z \subseteq \mathcal{X} \times \mathcal{Y}$, and outputs $\gamma_{1:M-1}$ so that (5) holds with probability at least $1 - \delta$. It iteratively chooses the thresholds from $\gamma_1$ to $\gamma_{M-1}$; at each step, it chooses $\gamma_m$ as small as possible subject to $p_{\text{err}} \leq \xi$. Note that $\gamma_m$ implicitly appears in $p_{\text{err}}$ in the constraint due to the dependence of $d_m(x)$ on $\gamma_m$. The challenge is enforcing the constraint since we cannot evaluate it. To this end, let

$$e_m := \mathbb{P}_{(x,y) \sim D} [\hat{y}_m(x) \neq y \wedge \hat{y}_m(x) \neq \hat{y}_M(x) \wedge d_m(x) = 1]$$
$$e'_m := \mathbb{P}_{(x,y) \sim D} [\hat{y}_M(x) \neq y \wedge \hat{y}_m(x) \neq \hat{y}_M(x) \wedge d_m(x) = 1],$$

then it is possible to show that $p_{\text{err}} = \sum_{m=1}^{M-1} e_m - e'_m$ (see proof of Theorem 2 in Appendix C). Then, we can compute bounds on $e_m$ and $e'_m$ using the following:

$$\mathbb{P}[\hat{y}_m(x) = y \mid \hat{y}_m(x) \neq \hat{y}_M(x) \wedge d_m(x) = 1] \in [\underline{c}_m, \bar{c}_m] := \hat{C}_0 \left( \hat{f}_m, Z_m, \frac{\delta}{3(M-1)} \right)$$

$$\mathbb{P}[\hat{y}_M(x) = y \mid \hat{y}_m(x) \neq \hat{y}_M(x) \wedge d_m(x) = 1] \in [\underline{c}'_m, \bar{c}'_m] := \hat{C}_0 \left( \hat{f}_M, Z_m, \frac{\delta}{3(M-1)} \right)$$

$$\mathbb{P}[\hat{y}_m(x) \neq \hat{y}_M(x) \wedge d_m(x) = 1] \in [\underline{r}_m, \bar{r}_m] := \hat{\Theta}_{\text{CP}} \left( W_m; \frac{\delta}{3(M-1)} \right),$$

where

$$Z_m := \{(x, y) \in Z \mid \hat{y}_m(x) \neq \hat{y}_M(x) \wedge d_m(x) = 1\}$$
$$W_m := \{\mathbb{1}(\hat{y}_m(x) \neq \hat{y}_M(x) \wedge d_m(x) = 1) \mid (x, y) \in Z\}.$$

Thus, we have $e_m \leq \bar{c}_m \bar{r}_m$ and $e'_m \geq \underline{c}'_m \underline{r}_m$, in which case it suffices to sequentially solve

$$\gamma_m = \underset{\gamma \in [0,1]}{\arg\min} \; \gamma \qquad \text{subj. to} \qquad \sum_{i=1}^{m} \bar{c}_i \bar{r}_i - \underline{c}'_i \underline{r}_i \leq \xi. \tag{6}$$

Here, $\bar{c}_m$, $\bar{r}_m$, $\underline{c}_m$, and $\underline{r}_m$ are implicitly a function of $\gamma$, which we can optimize using line search. We have the following guarantee; see Appendix C for a proof:

**Theorem 2** *We have $p_{err} \le \xi$ with probability at least $1 - \delta$ over $Z \sim D^n$.*

Moreover, the proposed greedy algorithm (6) is actually optimal in reducing inference speed when $M = 2$. Intuitively, we are always better off in terms of inference time by classifying more examples using a faster model. In particular, we have the following theorem; see Appendix D for a proof:

**Theorem 3** *If $M = 2$, $\gamma_1^*$ minimizes (6), and the classifiers $\hat{y}_m$ are faster for smaller $m$, then the resulting $\hat{y}_C$ is the fastest cascading classifier among cascading classifiers that satisfy $p_{err} \le \xi$.*

## 4 APPLICATION TO SAFE PLANNING

Robots acting in open world environments must rely on deep learning for tasks such as object recognition—*e.g.,* detect objects in a camera image; providing guarantees on these predictions is critical for safe planning. Safety requires not just that the robot is safe while taking the action, but that it can safely come to a stop afterwards—*e.g.,* that a robot can safely come to a stop before running into a wall. We consider a binary classification DNN trained to predict a probability $\hat{f}(x) \in [0, 1]$ of whether the robot is unsafe in this sense.[1] If $\hat{f}(x) \ge \gamma$ for some threshold $\gamma \in [0, 1]$, then the robot comes to a stop (*e.g.,* to ask a human for help). If the label $\mathbb{1}(\hat{f}(x) \ge \gamma)$ correctly predicts safety, then this policy ensures safety as long as the robot starts from a safe state (Li & Bastani, 2020). We apply our approach to choose $\gamma$ to ensure safety with high probability.

**Problem setup.** Given a performant but potentially unsafe policy $\hat{\pi}$ (*e.g.,* a DNN policy trained to navigate to the goal), our goal is to override $\hat{\pi}$ as needed to ensure safety. We assume that $\hat{\pi}$ is trained in a simulator, and our goal is to ensure that $\hat{\pi}$ is safe according to our model of the environment, which is already a challenging problem when $\hat{\pi}$ is a deep neural network over perceptual inputs. In particular, we do not address the sim-to-real problem.

Let $x \in \mathcal{X}$ be the states, $\mathcal{X}_{\text{safe}} \subseteq \mathcal{X}$ be the safe states (*i.e.,* our goal is to ensure the robot stays in $\mathcal{X}_{\text{safe}}$), $o \in \mathcal{O}$ be the observations, $u \in \mathcal{U}$ be the actions, $g : \mathcal{X} \times \mathcal{U} \to \mathcal{X}$ be the (deterministic) dynamics, and $h : \mathcal{X} \to \mathcal{O}$ be the observation function. A state $x$ is *recoverable* (denoted $x \in \mathcal{X}_{\text{rec}}$) if the robot can use $\hat{\pi}$ in state $x$ and then safely come to a stop using a *backup policy* $\pi_0$ (*e.g.,* braking).

Then, the *shield policy* uses $\hat{\pi}$ if $x \in \mathcal{X}_{\text{rec}}$, and $\pi_0$ otherwise (Bastani, 2019). This policy guarantees safety as long as an initial state is recoverable—*i.e.,* $x_0 \in \mathcal{X}_{\text{rec}}$. The challenge is determining whether $x \in \mathcal{X}_{\text{rec}}$. When we observe $x$, we can use model-based simulation to perform this check. However, in many settings, we only have access to observations—*e.g.,* camera images or LIDAR scans—so this approach does not apply. Instead, we propose to train a DNN to predict recoverability—*i.e.,*

$$\hat{y}(o) := \begin{cases} 1 \text{ (``un-recoverable'')} & \text{if } \hat{f}(o) \ge \gamma \\ 0 \text{ (``recoverable'')} & \text{otherwise} \end{cases} \qquad \text{where} \qquad o = h(x),$$

with the goal that $\hat{y}(o) \approx y^*(x) := \mathbb{1}(x \notin \mathcal{X}_{\text{rec}})$, resulting in the following the shield policy $\pi_{\text{shield}}$:

$$\pi_{\text{shield}}(o) := \begin{cases} \hat{\pi}(o) & \text{if } \hat{y}(o) = 0 \\ \pi_0(o) & \text{otherwise.} \end{cases}$$

**Safety guarantee.** Our main goal is to choose $\gamma$ so that $\pi_{\text{shield}}$ ensures safety with high probability—*i.e.,* given $\xi \in \mathbb{R}_{>0}$ and any distribution $D$ over initial states $\mathcal{X}_0 \subseteq \mathcal{X}_{\text{rec}}$, we have

$$p_{\text{unsafe}} := \mathbb{P}_{\zeta \sim D_{\pi_{\text{shield}}}}[\zeta \not\subseteq \mathcal{X}_{\text{safe}}] \le \xi, \tag{7}$$

where $\zeta(x_0, \pi) := (x_0, x_1, \dots)$ is a rollout from $x_0 \sim D$ generated using $\pi$—*i.e.,* $x_{t+1} = g(x_t, \pi(h(x_t)))$.[2] We assume the rollout terminates either once the robot reaches its goal, or once it switches to $\pi_0$ and comes to a stop; in particular, the robot never switches from $\pi_0$ back to $\hat{\pi}$.

**Success rate.** To maximize the success rate (*i.e.,* the rate at which the robot achieves its goal), we need to minimize how often $\pi_{\text{shield}}$ switches to $\pi_0$, which corresponds to maximizing $\gamma$.

---

[1]Since $|\mathcal{Y}| = 2$, $\hat{f}$ can be represented as a map $\hat{f} : \mathcal{X} \to [0, 1]$; the second component is simply $1 - \hat{f}(x)$.

[2]We can handle infinitely long rollouts, but in practice rollouts will be finite (but possibly arbitrarily long). and $D_{\pi_{\text{shield}}}$ is an induced distribution over rollouts $\zeta(x_0, \pi_{\text{shield}})$.

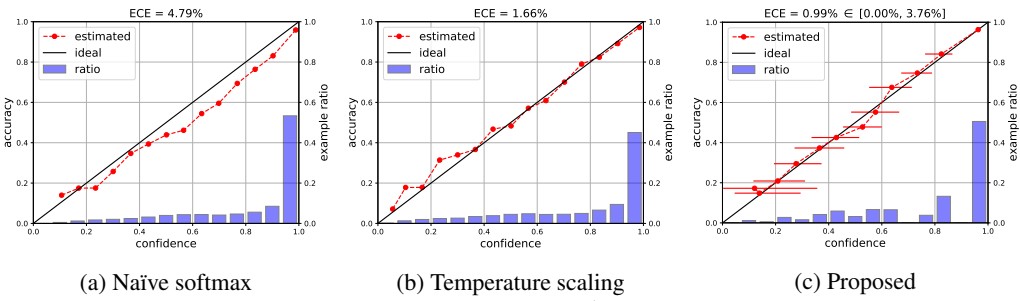

Figure 2: Calibration comparison; default parameters of $\hat{C}$ are $K = 20$, $n = 20{,}000$, and $\delta = 10^{-2}$.

**Algorithm.** Our algorithm takes the confidence predictor $\hat{f}$, desired bound $\xi \in \mathbb{R}_{>0}$ on the unsafety probability, confidence level $\delta \in [0,1]$, calibration set $W \subseteq \mathcal{X}^\infty$ of rollouts $\zeta \sim D_{\hat{\pi}}$, and calibration set $Z \subseteq \mathcal{O}$ of samples from distribution $\tilde{D}$ described below; see Appendix F.5 for details on sampling $\zeta$ and constructing $W$, $Z$, and $\tilde{D}$. We want to maximize $\gamma$ subject to $p_{\text{unsafe}} \leq \xi$, where $p_{\text{unsafe}}$ is implicitly a function of $\gamma$. However, we cannot evaluate $p_{\text{unsafe}}$, so we need an upper bound.

To this end, consider a rollout that the first unsafe state is encountered on step $t$ (*i.e.*, $x_t \notin \mathcal{X}_{\text{safe}}$ but $x_i \in \mathcal{X}_{\text{safe}}$ for all $i < t$), which we call an *unsafe rollout*, and denote the event that the unsafe rollout is encountered by $E_t$; we exploit the unsafe rollouts to bound $p_{\text{unsafe}}$. In particular, let $p_t := \mathbb{P}_{\zeta \sim D_{\hat{\pi}}}[E_t]$, and let $\bar{p} := \sum_{t=0}^\infty p_t$ be the probability that a rollout is unsafe. Then, consider a new distribution $\tilde{D}$ over $\mathcal{O}$ with a probability density function $p_{\tilde{D}}(o) := \sum_{t=0}^\infty p_{D_{\hat{\pi}}}(o \mid E_t) \cdot p_t / \bar{p}$, where $p_{D_{\hat{\pi}}}$ is the original probability density function of $D_{\hat{\pi}}$; in particular, we can draw an observation $o \sim \tilde{D}$ by sampling the observation of the first unsafe state from a rollout sample (and rejecting if the entire rollout is safe). Then, we can show the following (see a proof of Theorem 4 in Appendix E):

$$p_{\text{unsafe}} \leq \mathbb{P}_{o \sim \tilde{D}}[\hat{y}(o) = 0] \cdot \bar{p} =: \bar{p}_{\text{unsafe}}. \tag{8}$$

We use our confidence coverage predictor $\hat{C}_0$ to compute bounds on $\bar{p}_{\text{unsafe}}$—*i.e.*,

$$\mathbb{P}_{o \sim \tilde{D}}[\hat{y}(o) = 1] \in [\underline{c}, \bar{c}] := \hat{C}_0\left(\hat{f}, Z', \frac{\delta}{2}\right) \qquad \text{where} \qquad Z' := \{(o, 1) \mid o \in Z\},$$

$$\bar{p} \in [\underline{r}, \bar{r}] := \hat{\Theta}_{\text{CP}}\left(W', \frac{\delta}{2}\right) \qquad \text{where} \qquad W' := \{\mathbb{1}(\zeta \not\subseteq \mathcal{X}_{\text{safe}}) \mid \zeta \in W\}.$$

Here, $Z'$ is a labeled version of $Z$, where "1" denotes "un-recoverable", $n := |W|$, and $n' := |Z|$, where $n \geq n'$. Then, we have $\bar{p}_{\text{unsafe}} \leq \bar{r} \cdot (1 - \underline{c})$, so it suffices to solve the following problem:

$$\gamma := \underset{\gamma' \in [0,1]}{\arg\max} \ \gamma' \qquad \text{subj. to} \qquad \bar{r} \cdot (1 - \underline{c}) \leq \xi \tag{9}$$

Here, $\underline{c}$ is implicitly a function of $\gamma'$; thus, we use line search to solve this optimization problem. We have the following safety guarantee, see Appendix E for a proof:

**Theorem 4** *We have $p_{\text{unsafe}} \leq \xi$ with probability at least $1 - \delta$ over $W \sim D_{\hat{\pi}}^n$ and $Z \sim \tilde{D}^{n'}$.*

## 5 EXPERIMENTS

We demonstrate that how our proposed approach can be used to obtain provable guarantees in our two applications: fast DNN inference and safe planning. Additional results are in Appendix G.

### 5.1 CALIBRATION

We illustrate the calibration properties of our approach using reliability diagrams, which show the empirical accuracy of each bin as a function of the predicted confidence (Guo et al., 2017). Ideally, the accuracy should equal the predicted confidence, so the ideal curve is the line $y = x$. To draw our predicted confidence intervals in these plots, we need to rescale them; see Appendix F.3.

**Setup.** We use the ImageNet dataset (Russakovsky et al., 2015) and ResNet101 (He et al., 2016) for evaluation. We split the ImageNet validation set into $20{,}000$ calibration and $10{,}000$ test images.

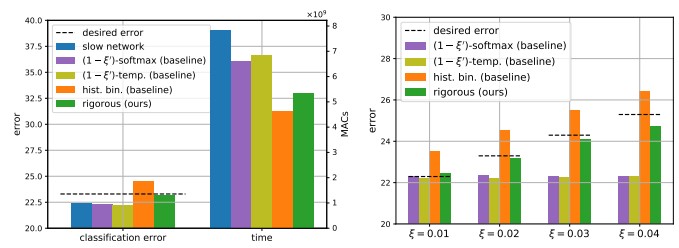

(a) Comparison to baselines.  (b) Varying desired error rate $\xi$.

| method | top-1 error (%) | CPU ($\mu$s) | GPU ($\mu$s) | MACs |
|---|---|---|---|---|
| slow network | 22.32 | 214.31 | 2.95 | $7.83 \times 10^9$ |
| $(1 - \xi')$-softmax (baseline) | 22.34 | 104.24 | 1.78 | $6.60 \times 10^9$ |
| $(1 - \xi')$-temperature scaling (baseline) | 22.22 | 109.57 | 1.81 | $6.83 \times 10^9$ |
| histogram binning (baseline) | 24.54 | 70.72 | 1.19 | $4.62 \times 10^9$ |
| rigorous (ours) | 23.26 | 85.58 | 1.34 | $5.33 \times 10^9$ |

(c) Running time comparison.

Figure 3: Fast DNN inference results; default parameters are $n = 20,000$, $\xi = 0.02$, and $\delta = 10^{-3}$.

**Baselines.** We consider three baselines: (i) naïve softmax of $\hat{f}$, (ii) temperature scaling (Guo et al., 2017), and (iii) histogram binning (Zadrozny & Elkan, 2001); see Appendix F.2 for details. For histogram binning and our approach, we use $K = 20$ bins of the same size.

**Metric.** We use expected calibration error (ECE) and reliability diagrams (see Appendix F.3).

**Results.** Results are shown in Figure 2. The ECE of the naïve softmax baseline is $4.79\%$ (Figure 2a), of temperature scaling enhances this to $1.66\%$ (Figure 2b), and of histogram binning is $0.99\%$ (Figure 2c). Our approach predicts an interval that include the empirical accuracy in all bins (solid red lines in Figure 2c); furthermore, the upper/lower bounds of the ECE over values in our bins is $[0.0\%, 3.76\%]$, which includes zero ECE. See Appendix G.1 for additional results.

## 5.2 FAST DNN INFERENCE

**Setup.** We use the same ImageNet setup along with ResNet101 as the calibration task. For the cascading classifier, we use the original ResNet101 as the slow network, and add a single exit branch (*i.e.*, $M = 2$) at a quarter of the way from the input layer. We train the newly added branch using the standard procedure for training ResNet101.

**Baselines.** We compare to naïve softmax and to calibrated prediction via temperature scaling, both using a threshold $\gamma_1 = 1 - \xi'$, where $\xi'$ is the sum of $\xi$ and the validation error of the slow model; intuitively, this threshold is the one we would use if the predicted probabilities are perfectly calibrated. We also compare to histogram binning—*i.e.*, our approach but using the means of each bin instead of the upper/lower bounds. See Appendix F.2 for details.

**Metrics.** First, we measure test set top-1 classification error (*i.e.*, $1 - $ accuracy), which we want to guarantee this lower than a desired error (*i.e.*, the error of the slow model and desired relative error $\xi$). To measure inference time, we consider the average number of multiplication-accumulation operations (MACs) used in inference per example. Note that the MACs are averaged over all examples in the test set since the combined model may use different numbers of MACs for different examples.

**Results.** The comparison results with the baselines are shown in Figure 3a. The original neural network model is denoted by "slow network", our approach (6) by "rigorous", and our baseline by "$(1 - \xi')$-softmax", "$(1 - \xi')$-temp.", and "hist. bin.". For each method, we plot the classification error and time in MACs. The desired error upper bound is plotted as a dotted line; the goal is for the classification error to be lower than this line. As can be seen, our method is guaranteed to achieve the desired error, while improving the inference time by $32\%$ compared to the slow model. On the other hand, the histogram baseline improves the inference time but fails to satisfy the desired error. Asymptotically, histogram binning is guaranteed to be perfectly calibrated, but it makes mistakes due to finite sample errors. The other baselines do not improve inference time. Next, Figure 3b shows the classification error as we vary the desired relative error $\xi$; our approach always achieves the desired error on the test set, and is often very close (which maximizes speed). However, the

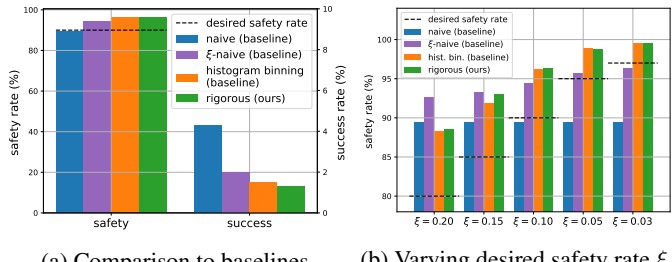

(a) Comparison to baselines.   (b) Varying desired safety rate $\xi$.

Figure 4: Safe planning results; default parameters are: $n = 20,000, \xi = 0.1, \delta = 10^{-2}$.

baselines often violate the desired error bound. Finally, the MACs metric is only an approximation of the actual inference time. To complement MACs, we also measure CPU and GPU time using the PyTorch profiler. In Figure 3c, we show the inference times for each method, where trends are as before; our approach improves running time by 54%, while only reducing classification error by 1%. The histogram baseline is faster than our approach, but does not satisfy the error guarantee. These results include the time needed to compute the intervals, which is negligible.

## 5.3  SAFE PLANNING

**Setup.** We evaluate on AI Habitat (Savva et al., 2019), an indoor robot simulator that provides agents with observations $o = h(x)$ that are RGB camera images. The safety constraint is to avoid colliding with obstacles such as furniture in the environment. We use the learned policy $\hat{\pi}$ available with the environment. Then, we train a recoverability predictor, trained using 500 rollouts with a horizon of 100. We calibrate this model on an additional $n$ rollouts.

**Baselines.** We compare to three baselines: (i) histogram binning—*i.e.,* our approach but using the means of each bin rather than upper/lower bounds, (ii) a naïve approach of choosing $\gamma = 0.5$, and (iii) a naïve but adaptive approach of choosing $\gamma = \xi$, called "$\xi$-naïve".

**Metrics.** We measure both the safety rate and the success rate; in particular, a rollout is successful if the robot reaches its goal state, and a rollout is safe if it does not reach any unsafe states.

**Results.** We show results in Figure 4a. The desire safety rate $\xi$ is shown by the dotted line—*i.e.,* we expect the safety rate to be above this line. As can be seen, our approach achieves the desired safety rate. While it sacrifices success rate, this is because the underlying learned policy $\hat{\pi}$ is frequently unsafe; in particular, it is only safe in about 30% of rollouts. The naïve approach fails to satisfy the safety constraint. The $\xi$-naïve approach tends to be optimistic, and also fails when $\xi = 0.03$ (Figure 4b). The histogram baseline performs similarly to our approach. The main benefit of our approach is providing the absolute guarantee on safety, which the histogram baseline does not provide. Thus, in this case, our approach can provide this guarantee while achieving similar performance. Figure 4b shows the safety rate as we vary the desired safety rate $\xi$. Both our approach and the baseline satisfy the desired safety guarantee, whereas the naive approaches do not always do so.

## 6  CONCLUSION

We have proposed a novel algorithm for calibrated prediction that provides PAC guarantees, and demonstrated how our approach can be applied to fast DNN inference and safe planning. There are many directions for future work—*e.g.,* leveraging these techniques in more application domains and extending our approach to settings with distribution shift (see Appendix F.1 for a discussion).

ACKNOWLEDGMENTS

This work was supported in part by AFRL/DARPA FA8750-18-C-0090, ARO W911NF-20-1-0080, DARPA FA8750-19-2-0201, and NSF CCF 1910769. Any opinions, findings and conclusions or recommendations expressed in this material are those of the authors and do not necessarily reflect the views of the Air Force Research Laboratory (AFRL), the Army Research Office (ARO), the Defense Advanced Research Projects Agency (DARPA), or the Department of Defense, or the United States Government.

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

## A    CONNECTION TO PAC LEARNING THEORY

We explain the connection to PAC learning theory. First, note that we can represent $\hat{C}$ as a confidence interval around the empirical estimate of $c_{\hat{f}}(x)$—i.e., $\hat{c}_{\hat{f}}(x) := \sum_{(x',y')\in S_x} \mathbb{1}(y' = \hat{y}(x'))/|S_x|$, where $S_x = \{(x', y') \mid \hat{p}(x') \in B_{\kappa_{\hat{f}}(x)}\}$. Then, we can write

$$\hat{C}(x) = [\hat{c}_{\hat{f}}(x) - \underline{\epsilon}_x, \hat{c}_{\hat{f}}(x) + \bar{\epsilon}_x].$$

In this case, (3) is equivalent to

$$\mathbb{P}_{Z\sim D^n}\left[\bigwedge_{x\in\mathcal{X}} \hat{c}_{\hat{f}}(x) - \underline{\epsilon}_{\kappa_{\hat{f}}(x)} \le c_{\hat{f}}(x) \le \hat{c}_{\hat{f}}(x) + \bar{\epsilon}_{\kappa_{\hat{f}}(x)}\right] \ge 1 - \delta, \tag{10}$$

for some $\underline{\epsilon}_1, \bar{\epsilon}_1, \ldots, \underline{\epsilon}_K, \bar{\epsilon}_K$. In this bound, "approximately" refers to the fact that the empirical estimate $\hat{c}_{\hat{f}}(x)$ is within $\epsilon$ of the true value $c_{\hat{f}}(x)$, and "probably" refers to the fact that this error bound holds with high probability over the training data $Z \sim D^n$. By abuse of terminology, we refer to the confidence interval predictor $\hat{C}$ as PAC rather than just $\hat{c}_{\hat{f}}(x)$.

Alternatively, we also have the following connection to PAC learning theory:

**Definition 3**  Given $\epsilon, \delta \in \mathbb{R}_{>0}$ and $n \in \mathbb{N}$, $\hat{C}$ is *probably approximately correct (PAC)* if, for any distribution $D$, we have

$$\mathbb{P}_{Z\sim D^n}\left[\mathbb{P}_{x\sim D}\left[c_{\hat{f}}(x) \in \hat{C}(x; \hat{f}, Z)\right] \ge 1 - \epsilon\right] \ge 1 - \delta. \tag{11}$$

The following theorem shows that the proposed confidence coverage predictor $\hat{C}$ satisfies the PAC guarantee in Definition 3.

**Theorem 5** *Our confidence coverage predictor $\hat{C}$ satisfies Definition 3 for all $\epsilon, \delta \in \mathbb{R}_{>0}$, and $n \in \mathbb{N}$.*

*Proof.* We exploit the independence of each bin for the proof. Let $\theta_{\kappa_{\hat{f}}(x)} := c_{\hat{f}}(x)$, which is the parameter of the Binomial distribution of the $\kappa_{\hat{f}}(x)$th bin, the following holds:

$$
\mathop{\mathbb{P}}_{Z \sim D^n} \left[ \mathop{\mathbb{P}}_{x \sim D} \left[ c_{\hat{f}}(x) \in \hat{C}(x; \hat{f}, Z, \delta) \right] \geq 1 - \epsilon \right]
$$

$$
= \mathop{\mathbb{P}}_{Z \sim D^n} \left[ \mathop{\mathbb{P}}_{x \sim D} \left[ c_{\hat{f}}(x) \in \hat{C}(x; \hat{f}, Z, \delta) \wedge \bigvee_{k=1}^{K} \hat{p}(x) \in B_k \right] \geq 1 - \epsilon \right]
$$

$$
= \mathop{\mathbb{P}}_{Z \sim D^n} \left[ \sum_{k=1}^{K} \mathop{\mathbb{P}}_{x \sim D} \left[ c_{\hat{f}}(x) \in \hat{C}(x; \hat{f}, Z, \delta) \wedge \hat{p}(x) \in B_k \right] \geq 1 - \epsilon \right]
$$

$$
= \mathop{\mathbb{P}}_{Z \sim D^n} \left[ \sum_{k=1}^{K} \mathop{\mathbb{P}}_{x \sim D} \left[ c_{\hat{f}}(x) \in \hat{C}(x; \hat{f}, Z, \delta) \,\Big|\, \hat{p}(x) \in B_k \right] \mathop{\mathbb{P}}_{x \sim D} [\hat{p}(x) \in B_k] \geq 1 - \epsilon \right]
$$

$$
= \mathop{\mathbb{P}}_{Z \sim D^n} \left[ \sum_{k=1}^{K} \mathop{\mathbb{P}}_{x \sim D} \left[ \theta_k \in \hat{\Theta}_{\mathrm{CP}} \left( W_k; \frac{\delta}{K} \right) \,\Big|\, \hat{p}(x) \in B_k \right] \mathop{\mathbb{P}}_{x \sim D} [\hat{p}(x) \in B_k] \geq 1 - \epsilon \right]
$$

$$
= \mathop{\mathbb{P}}_{Z \sim D^n} \left[ \sum_{k=1}^{K} \mathbb{1} \left[ \theta_k \in \hat{\Theta}_{\mathrm{CP}} \left( W_k; \frac{\delta}{K} \right) \right] \mathop{\mathbb{P}}_{x \sim D} [\hat{p}(x) \in B_k] \geq 1 - \epsilon \right]
$$

$$
\geq \mathop{\mathbb{P}}_{Z \sim D^n} \left[ \sum_{k=1}^{K} \mathbb{1} \left[ \theta_k \in \hat{\Theta}_{\mathrm{CP}} \left( W_k; \frac{\delta}{K} \right) \right] \mathop{\mathbb{P}}_{x \sim D} [\hat{p}(x) \in B_k] \geq 1 - \epsilon \right.
$$
$$
\left. \wedge \bigwedge_{k=1}^{K} \mathbb{1} \left[ \theta_k \in \hat{\Theta}_{\mathrm{CP}} \left( W_k; \frac{\delta}{K} \right) \right] = 1 \right]
$$

$$
= \mathop{\mathbb{P}}_{Z \sim D^n} \left[ \sum_{k=1}^{K} \mathbb{1} \left[ \theta_k \in \hat{\Theta}_{\mathrm{CP}} \left( W_k; \frac{\delta}{K} \right) \right] \mathop{\mathbb{P}}_{x \sim D} [\hat{p}(x) \in B_k] \geq 1 - \epsilon \,\Bigg|\, \bigwedge_{k=1}^{K} \mathbb{1} \left[ \theta_k \in \hat{\Theta}_{\mathrm{CP}} \left( W_k; \frac{\delta}{K} \right) \right] = 1 \right]
$$
$$
\mathop{\mathbb{P}}_{Z \sim D^n} \left[ \bigwedge_{k=1}^{K} \mathbb{1} \left[ \theta_k \in \hat{\Theta}_{\mathrm{CP}} \left( W_k; \frac{\delta}{K} \right) \right] = 1 \right]
$$

$$
= \mathop{\mathbb{P}}_{Z \sim D^n} \left[ \sum_{k=1}^{K} \mathop{\mathbb{P}}_{x \sim D} [\hat{p}(x) \in B_k] \geq 1 - \epsilon \,\Bigg|\, \bigwedge_{k=1}^{K} \mathbb{1} \left[ \theta_k \in \hat{\Theta}_{\mathrm{CP}} \left( W_k; \frac{\delta}{K} \right) \right] = 1 \right]
$$
$$
\mathop{\mathbb{P}}_{Z \sim D^n} \left[ \bigwedge_{k=1}^{K} \mathbb{1} \left[ \theta_k \in \hat{\Theta}_{\mathrm{CP}} \left( W_k; \frac{\delta}{K} \right) \right] = 1 \right]
$$

$$
= \mathop{\mathbb{P}}_{Z \sim D^n} \left[ 1 \geq 1 - \epsilon \,\Bigg|\, \bigwedge_{k=1}^{K} \mathbb{1} \left[ \theta_k \in \hat{\Theta}_{\mathrm{CP}} \left( W_k; \frac{\delta}{K} \right) \right] = 1 \right] \mathop{\mathbb{P}}_{Z \sim D^n} \left[ \bigwedge_{k=1}^{K} \mathbb{1} \left[ \theta_k \in \hat{\Theta}_{\mathrm{CP}} \left( W_k; \frac{\delta}{K} \right) \right] = 1 \right]
$$

$$
= \mathop{\mathbb{P}}_{Z \sim D^n} \left[ \bigwedge_{k=1}^{K} \mathbb{1} \left[ \theta_k \in \hat{\Theta}_{\mathrm{CP}} \left( W_k; \frac{\delta}{K} \right) \right] = 1 \right]
$$

$$
= \mathop{\mathbb{P}}_{Z \sim D^n} \left[ \bigwedge_{k=1}^{K} \theta_k \in \hat{\Theta}_{\mathrm{CP}} \left( W_k; \frac{\delta}{K} \right) \right]
$$

$$
\geq 1 - \delta,
$$

where the last inequality holds by the union bound. $\qquad\square$

# B PROOF OF THEOREM 1

We prove this by exploiting the independence of each bin. Recall that $\hat{C}(x) := [\hat{c}_{\hat{f}}(x) - \underline{\epsilon}_x, \hat{c}_{\hat{f}}(x) + \bar{\epsilon}_x]$, and the interval is obtained by applying the Clopper-Pearson interval with confidence level $\frac{\delta}{K}$ at each bin. Then, the following holds due the Clopper-Pearson interval for all $k \in \{1, 2, \dots, K\}$:

$$\mathbb{P}\left[|c_{\hat{f},k} - \hat{c}_{\hat{f},k}| > \epsilon_k\right] \leq \frac{\delta}{K}$$

where $c_{\hat{f},k} := c_{\hat{f}}(x)$ and $\hat{c}_{\hat{f},k} := \hat{c}_{\hat{f}}(x)$ for $x$ such that $\kappa_{\hat{f}}(x) = k$, and $\epsilon_k := \max(\underline{\epsilon}_x, \bar{\epsilon}_x)$. By applying the union bound, the following also holds:

$$\mathbb{P}\left[\bigwedge_{k=1}^{K} |c_{\hat{f},k} - \hat{c}_{\hat{f},k}| > \epsilon_k\right] \leq \delta,$$

Considering the fact that $\mathcal{X}$ is partitioned into $K$ spaces due to the binning and the equivalence form (10) of the PAC criterion in Definition 3, the claimed statement holds.

# C PROOF OF THEOREM 2

We drop probabilities over $(x, y) \sim D$. First, we decompose the error of a cascading classifier $\mathbb{P}\left[\hat{y}_C(x) \neq y\right]$ as follows:

$$\begin{aligned}
\mathbb{P}\left[\hat{y}_C(x) \neq y\right] &= \mathbb{P}\left[\hat{y}_C(x) \neq y \wedge \left(\hat{y}_C(x) = \hat{y}_M(x) \vee \hat{y}_C(x) \neq \hat{y}_M(x)\right)\right] \\
&= \mathbb{P}\left[\left(\hat{y}_C(x) \neq y \wedge \hat{y}_C(x) = \hat{y}_M(x)\right) \vee \left(\hat{y}_C(x) \neq y \wedge \hat{y}_C(x) \neq \hat{y}_M(x)\right)\right] \\
&= \mathbb{P}\left[\hat{y}_C(x) \neq y \wedge \hat{y}_C(x) = \hat{y}_M(x)\right] + \mathbb{P}\left[\hat{y}_C(x) \neq y \wedge \hat{y}_C(x) \neq \hat{y}_M(x)\right],
\end{aligned}$$

where the last equality holds since the events of $\hat{y}_C(x) = \hat{y}_M(x)$ and of $\hat{y}_C(x) \neq \hat{y}_M(x)$ are disjoint. Similarly, for the error of a slow classifier $\mathbb{P}\left[\hat{y}_M(x) \neq y\right]$, we have:

$$\mathbb{P}\left[\hat{y}_M(x) \neq y\right] = \mathbb{P}\left[\hat{y}_M(x) \neq y \wedge \hat{y}_C(x) = \hat{y}_M(x)\right] + \mathbb{P}\left[\hat{y}_M(x) \neq y \wedge \hat{y}_C(x) \neq \hat{y}_M(x)\right].$$

Thus, the error difference can be represented as follows:

$$\begin{aligned}
\mathbb{P}\left[\hat{y}_C(x) \neq y\right] &- \mathbb{P}\left[\hat{y}_M(x) \neq y\right] \\
&= \mathbb{P}\left[\hat{y}_C(x) \neq y \wedge \hat{y}_C(x) \neq \hat{y}_M(x)\right] - \mathbb{P}\left[\hat{y}_M(x) \neq y \wedge \hat{y}_C(x) \neq \hat{y}_M(x)\right]. \quad (12)
\end{aligned}$$

To complete the proof, we need to upper bound (12) by $\xi$. Define the following events:

$$\begin{aligned}
D_m &:= \bigwedge_{i=1}^{m-1} (\hat{p}_i(x) < \gamma_i) \wedge \hat{p}_m(x) \geq \gamma_m \qquad (\forall m \in \{1, ..., M-1\}) \\
D_M &:= \bigwedge_{i=1}^{M-1} (\hat{p}_i(x) < \gamma_i) \\
E_C &:= \hat{y}_C(x) \neq \hat{y}_M(x) \\
E_m &:= \hat{y}_m(x) \neq \hat{y}_M(x) \qquad (\forall m \in \{1, ..., M-1\}) \\
F_C &:= \hat{y}_C(x) \neq y \\
F_m &:= \hat{y}_m(x) \neq y \qquad (\forall m \in \{1, ..., M-1\}) \\
G &:= \hat{y}_M(x) \neq y,
\end{aligned}$$

where $D_1, D_2, \ldots, D_M$ form a partition of a sample space. Then, we have:

$$\mathbb{P}\left[\hat{y}_C(x) \neq y \wedge \hat{y}_C(x) \neq \hat{y}_M(x)\right] = \mathbb{P}\left[F_C \wedge E_C\right]$$

$$= \mathbb{P}\left[F_C \wedge E_C \wedge \bigvee_{m=1}^{M} D_m\right]$$

$$= \mathbb{P}\left[\bigvee_{m=1}^{M} \left(F_C \wedge E_C \wedge D_m\right)\right]$$

$$= \sum_{m=1}^{M} \mathbb{P}\left[F_C \wedge E_C \wedge D_m\right]$$

$$= \sum_{m=1}^{M} \mathbb{P}\left[F_m \wedge E_m \wedge D_m\right]$$

$$= \sum_{m=1}^{M} \mathbb{P}\left[F_m \mid E_m \wedge D_m\right] \cdot \mathbb{P}\left[E_m \wedge D_m\right],$$

Similarly, we have:

$$\mathbb{P}\left[\hat{y}_M(x) \neq y \wedge \hat{y}_C(x) \neq \hat{y}_M(x)\right] = \sum_{m=1}^{M} \mathbb{P}\left[G \mid E_m \wedge D_m\right] \cdot \mathbb{P}\left[E_m \wedge D_m\right].$$

Thus, (12) can be rewritten as follows:

$$\mathbb{P}\left[\hat{y}_C(x) \neq y\right] - \mathbb{P}\left[\hat{y}_M(x) \neq y\right]$$

$$= \sum_{m=1}^{M} \left(\mathbb{P}\left[F_m \mid E_m \wedge D_m\right] \cdot \mathbb{P}\left[E_m \wedge D_m\right] - \mathbb{P}\left[G \mid E_m \wedge D_m\right] \cdot \mathbb{P}\left[E_m \wedge D_m\right]\right)$$

$$= \sum_{m=1}^{M} \left(e_m - e'_m\right)$$

$$= \sum_{m=1}^{M-1} \left(e_m - e'_m\right)$$

$$\leq \xi,$$

where the last equality holds since $e_M - e'_M = 0$, and the last inequality holds due to (6) with probability at least $1 - \delta$, thus proves the claim.

## D    PROOF OF THEOREM 3

Suppose there is $\gamma'_1$ which is different to $\gamma_1^*$ and produces a faster cascading classifier than the cascading classifier with $\gamma_1^*$. Since $\gamma_1^*$ is the optimal solution of (6), $\gamma'_1 > \gamma_1^*$. This further implies that the less number of examples exits at the first branch of the cascading classifier with $\gamma'_1$, but these examples are classified by the upper, slower branch. This means that the overall inference speed of the cascading classifier with $\gamma'_1$ is slower then that with $\gamma_1^*$, which leads to a contradiction.

## E    PROOF OF THEOREM 4

For clarity, we use $r$ to denote a state $x$ is "recoverable" (*i.e.*, $y^*(x) = 0$) and $u$ to denote a state $x$ is "un-recoverable" (*i.e.*, $y^*(x) = 1$). Now, note that a rollout $\zeta(x_0, \pi_{\text{shield}}) := (x_0, x_1, \ldots)$ is unsafe if (i) at some step $t$, we have $y^*(x_t) = u$ (*i.e.*, $x_t$ is not recoverable), yet $\hat{y}(o_t) = r$, where $o_t = h(x_t)$ (*i.e.*, $\hat{y}$ predicts $x_t$ is recoverable), and furthermore (ii) for every step $i \leq t - 1$, $y^*(x_i) = \hat{y}(o_i) = r$—*i.e.*,

$$p_{\text{unsafe}} = \mathbb{P}_{\xi \sim D_{\pi_{\text{shield}}}}\left[\bigvee_{t=0}^{\infty}\left(\bigwedge_{i=0}^{t-1}\left(y^*(x_i) = r \wedge \hat{y}(o_i) = r\right) \wedge \left(y^*(x_t) = u \wedge \hat{y}(o_t) = r\right)\right)\right]. \quad (13)$$

Condition (i) is captured by the second parenthetical inside the probability; intuitively, it says that $\hat{y}(o_t)$ is a false negative. Condition (ii) is captured by the first parenthetical inside the probability; intuitively, it says that $\hat{y}(o_i)$ is a true negative for any $i \leq t - 1$. Next, let the event $E_t$ be

$$E_t := \bigwedge_{i=0}^{t-1} y^*(x_i) = r \wedge y^*(x_t) = u,$$

then the following holds:

$$\mathbb{P}_{\xi \sim D_{\pi_{\text{shield}}}} \left[ \bigvee_{t=0}^{\infty} \left( \bigwedge_{i=0}^{t-1} \left( y^*(x_i) = r \wedge \hat{y}(o_i) = r \right) \wedge \left( y^*(x_t) = u \wedge \hat{y}(o_t) = r \right) \right) \right]$$

$$= \mathbb{P}_{\xi \sim D_{\pi_{\text{shield}}}} \left[ \bigvee_{t=0}^{\infty} \left( \bigwedge_{i=0}^{t-1} \left( y^*(x_i) = r \wedge y^*(x_t) = u \right) \wedge \left( \bigwedge_{i=0}^{t-1} \hat{y}(o_i) = r \wedge \hat{y}(o_t) = r \right) \right) \right]$$

$$= \mathbb{P}_{\xi \sim D_{\pi_{\text{shield}}}} \left[ \bigvee_{t=0}^{\infty} \left( E_t \wedge \bigwedge_{i=0}^{t-1} \hat{y}(o_i) = r \wedge \hat{y}(o_t) = r \right) \right]$$

$$\leq \mathbb{P}_{\xi \sim D_{\pi_{\text{shield}}}} \left[ \bigvee_{t=0}^{\infty} \left( E_t \wedge \hat{y}(o_t) = r \right) \right].$$

Recall that $\bar{p} := \sum_{t=0}^{\infty} \mathbb{P}_{\xi \sim D_{\hat{\pi}}}[E_t]$ and $p_{\tilde{D}}(o) := \sum_{t=0}^{\infty} p_{D_{\hat{\pi}}}(o \mid E_t) \cdot \mathbb{P}_{\xi \sim D_{\hat{\pi}}}[E_t]/\bar{p}$; then we can upper-bound (13) as follows:

$$p_{\text{unsafe}} \leq \mathbb{P}_{\xi \sim D_{\pi_{\text{shield}}}} \left[ \bigvee_{t=0}^{\infty} \left( E_t \wedge \hat{y}(o_t) = r \right) \right]$$

$$= \sum_{t=0}^{\infty} \mathbb{P}_{\xi \sim D_{\pi_{\text{shield}}}} [E_t \wedge \hat{y}(o_t) = r]$$

$$= \sum_{t=0}^{\infty} \mathbb{P}_{\xi \sim D_{\pi_{\text{shield}}}} [\hat{y}(o_t) = r \mid E_t] \cdot \mathbb{P}_{\xi \sim D_{\pi_{\text{shield}}}} [E_t]$$

$$= \sum_{t=0}^{\infty} \mathbb{P}_{\xi \sim D_{\pi_{\text{shield}}}} [\hat{y}(o) = r \mid E_t] \cdot \mathbb{P}_{\xi \sim D_{\pi_{\text{shield}}}} [E_t]$$

$$= \sum_{t=0}^{\infty} \int \mathbb{1}(\hat{y}(o) = r) \cdot p_{D_{\pi_{\text{shield}}}}(o \mid E_t) \cdot \mathbb{P}_{\xi \sim D_{\pi_{\text{shield}}}} [E_t] \, \mathrm{d}o$$

$$= \int \mathbb{1}(\hat{y}(o) = r) \sum_{t=0}^{\infty} p_{D_{\pi_{\text{shield}}}}(o \mid E_t) \cdot \mathbb{P}_{\xi \sim D_{\pi_{\text{shield}}}} [E_t] \, \mathrm{d}o$$

$$\leq \int \mathbb{1}(\hat{y}(o) = r) \sum_{t=0}^{\infty} p_{D_{\hat{\pi}}}(o \mid E_t) \cdot \mathbb{P}_{\xi \sim D_{\hat{\pi}}} [E_t] \, \mathrm{d}o$$

$$= \int \mathbb{1}(\hat{y}(o) = r) p_{\tilde{D}}(o) \bar{p} \, \mathrm{d}o$$

$$= \bar{p} \cdot \mathbb{P}_{o \sim \tilde{D}} [\hat{y}(o) = r],$$

where we use the fact that $E_t$ are disjoint by construction for the first equality, and we use $o$ without time index $t$ for the third equality since it clearly represents the last observation if it is conditioned on $E_t$. Moreover, the last inequality holds due to the following: (i) $p_{D_{\pi_{\text{shield}}}}(o \mid E_t) = p_{D_{\hat{\pi}}}(o \mid E_t)$, since $E_t$ implies that the backup policy of $\pi_{\text{shield}}$ isn't activated up to the step $t$, so $\pi_{\text{shield}} = \hat{\pi}$, and (ii) $\mathbb{P}_{\xi \sim D_{\pi_{\text{shield}}}}[E_t] \leq \mathbb{P}_{\xi \sim D_{\hat{\pi}}}[E_t]$, since $\pi_{\text{shield}}$ is less likely to reach unsafe states by its design than $\hat{\pi}$. Thus, the constraint in (9) implies $p_{\text{unsafe}} \leq \xi$ with probability at least $1 - \delta$, so the claim follows.

# F  ADDITIONAL DISCUSSION

## F.1  LIMITATION TO ON-DISTRIBUTION SETTING

Our PAC guarantees (*i.e.,* Theorems 1 & 5) transfer to the test distribution if it is identical to the validation distribution. We believe that providing theoretical guarantees for out-of-distribution data is an important direction; however, we believe that our work is an important stepping stone towards this goal. In particular, to the best of our knowledge, we do not know of any existing work that provides theoretical guarantees on calibrated probabilities even for the in-distribution case. One possible direction is to use our approach in conjunction with covariate shift detectors—*e.g.,* (Gretton et al., 2012). Alternatively, it may be possible to directly incorporate ideas from recent work on calibrated prediction with covariate shift (Park et al., 2020b) or uncertainty set prediction with covariate shift (Cauchois et al., 2020; Tibshirani et al., 2019). In particular, we can use importance weighting $q(x)/p(x)$, where $p(x)$ is the training distribution and $q(x)$ is the test distribution, to reweight our training examples, enabling us to transfer our guarantees from the training set to the test distribution. The key challenge is when these weights are unknown. In this case, we can estimate them given a set of unlabeled examples from the test distribution (Park et al., 2020b), but we then need to account for the error in our estimates.

## F.2  BASELINES

The following includes brief descriptions on baselines that we used in experiments.

**Histogram binning.** This algorithm calibrates the top-label confidence prediction of $\hat{f}$ by sorting the calibration examples $(x, y)$ into bins $B_i$ based on their predicted top-label confidence—*i.e.,* $(x, y)$ is associated with $B_i$ if $\hat{p}(x) \in B_i$. Then, for each bin, it computes the empirical confidence $\hat{p}_i := \frac{1}{|S_i|} \sum_{(x,y) \in S_i} \mathbb{1}(\hat{y}(x) = y)$, where $S_i$ is the set of labeled examples that are associated with bin $B_i$—*i.e.,* the empirical counterpart of the true confidence in (2). Finally, during the test-time, it returns a predicted confidence $\hat{p}_i$ for all future test examples $x$ if $\hat{p}(x) \in B_i$.

$(1 - \xi')$**-softmax.** In fast DNN inference, a threshold can be heuristically chosen based on the desired relative error $\xi$ and the validation error of the slow model. In particular, when a cascading classifier consists of two branches—*i.e.,* $M = 2$, the threshold of the first branch is chosen by $\gamma_1 = 1 - \xi'$, where $\xi'$ is the sum of $\xi$ and the validation error of the slow model. We call this approach $(1 - \xi')$-softmax.

$(1 - \xi')$**-temperature scaling.** A more advanced approach is to first calibrate each branch to get better confidence. We consider using temperature scaling to do so—*i.e.,* we first calibrate each branch using the temperature scaling, and then use the branch threshold by $\gamma_1 = 1 - \xi'$ when $M = 2$. We call this approach $(1 - \xi')$-temperature scaling.

## F.3  CALIBRATION: INDUCED INTERVALS FOR ECE AND RELIABILITY DIAGRAM

**ECE.** The expected calibration error (ECE), which is one way to measure calibration performance, is defined as follows:

$$\text{ECE} := \sum_{j=1}^{J} \frac{|S_j|}{|S|} \left| \frac{1}{|S_j|} \sum_{(x,y) \in S_j} \hat{p}(x) - \frac{1}{|S_j|} \sum_{(x,y) \in S_j} \mathbb{1}(\hat{y}(x) = y) \right|,$$

where $J$ is the total number of bins for ECE, $S \subseteq \mathcal{X} \times \mathcal{Y}$ is the evaluation set, and $S_j$ is the set of labeled examples associated to the $j$th bin—*i.e.,* $(x, y) \in S_j$ if $\hat{p}(x) \in B_j$.

A confidence coverage predictor $\hat{C}(x)$ outputs an interval instead of a point estimate $\hat{p}(x)$ of the confidence. To evaluate the confidence coverage predictor, we remap intervals using the ECE formulation. In particular, we equivalently represents $\hat{C}(x)$ by a mean confidence $\hat{c}_{\hat{f}}(x)$ and differences from the mean—*i.e.,* $\hat{C}(x) = [\underline{c}(x), \overline{c}(x)] = [\hat{c}_{\hat{f}}(x) - \underline{\epsilon}_x, \hat{c}_{\hat{f}}(x) + \overline{\epsilon}_x]$ (see Appendix A for a description on this equivalent representation). Then, we sort each labeled example into

bins using $\hat{c}_{\hat{f}}(x)$ to form $S_j$. Next, we consider an interval instead of $\hat{p}(x)$ to compute ECE—
*i.e.,* $\text{ECE}_{\text{induced}} := \left[\underline{\text{ECE}}, \overline{\text{ECE}}\right]$, where

$$\underline{\text{ECE}} := \sum_{j=1}^{J} \frac{|S_j|}{|S|} \inf_{\hat{p}_j \in \text{Conf}_j} \left| \hat{p}_j - \frac{1}{|S_j|} \sum_{(x,y) \in S_j} \mathbb{1}(\hat{y}(x) = y) \right|,$$

$$\overline{\text{ECE}} := \sum_{j=1}^{J} \frac{|S_j|}{|S|} \sup_{\hat{p}_j \in \text{Conf}_j} \left| \hat{p}_j - \frac{1}{|S_j|} \sum_{(x,y) \in S_j} \mathbb{1}(\hat{y}(x) = y) \right|, \text{ and}$$

$$\text{Conf}_j := \left[\underline{\text{Conf}}_j, \overline{\text{Conf}}_j\right] := \left[ \min_{(x,y) \in S_j} \underline{c}(x), \max_{(x,y) \in S_j} \overline{c}(x) \right].$$

**Reliability diagram.** This evaluation technique is a pictorial summary of the ECE, where the $x$-axis represents the mean confidence $\frac{1}{|S_j|} \sum_{(x,y) \in S_j} \hat{p}(x)$ for each bin, and the $y$-axis represents the mean accuracy $\frac{1}{|S_j|} \sum_{(x,y) \in S_j} \mathbb{1}(\hat{y}(x) = y)$ for each bin. If an interval from a confidence coverage predictor is given, then the mean confidence is replaced by $\text{Conf}_j$, resulting in visualizing an interval instead of a point.

### F.4   FAST DNN INFERENCE: CASCADING CLASSIFIER TRAINING

We describe a way to train a cascading classifier with $M$ branches. Basically, we consider to independently train $M$ different neural networks with a shared backbone. In particular, we first train the $M$th branch using a training set by minimizing a cross-entropy loss. Then, we train the $(M-1)$th branch, which consists of two parts: a backbone part from the $M$th branch, and the head of the $(M-1)$th branch. Here, the backbone part is already trained in the previous stage, so we do not update the backbone part and only train the head of this branch using the *same* training set by minimizing the *same* cross-entropy loss (with the *same* optimization hyperparameters). This step is done repeatedly down to the first branch.

### F.5   SAFE PLANNING: DATA COLLECTION FROM A SIMULATOR

We collect required data from a simulator, where a given policy $\hat{\pi}$ is already learned over the simulator. We describe how we form the necessary data from rollouts sampled from the simulator.

First, to sample a rollout $\zeta$, we use $\hat{\pi}$ from a random initial state $x_0 \sim D$; we denote the sequence of states visited as a rollout $\zeta(x_0, \hat{\pi}) := (x_0, x_1, \dots)$. We denote the induced distribution over rollouts by $\zeta \sim D_{\hat{\pi}}$. Note that the $\zeta$ contains unsafe states since $\hat{\pi}$ is potentially unsafe. However, when constructing our recoverability classifier, we only use the sequence of safe states, followed by a single unsafe state. In particular, we let $W$ be a set of i.i.d. sampled rollouts $\zeta \sim D_{\hat{\pi}}$. Next, for a given rollout, we consider the observation in the first unsafe state in that rollout (if one exists); we denote the distribution over such observations by $\tilde{D}$. Finally, we take $Z$ to be a set of sampled observations $o \sim \tilde{D}$.

## G   ADDITIONAL EXPERIMENTS

### G.1   CALIBRATION

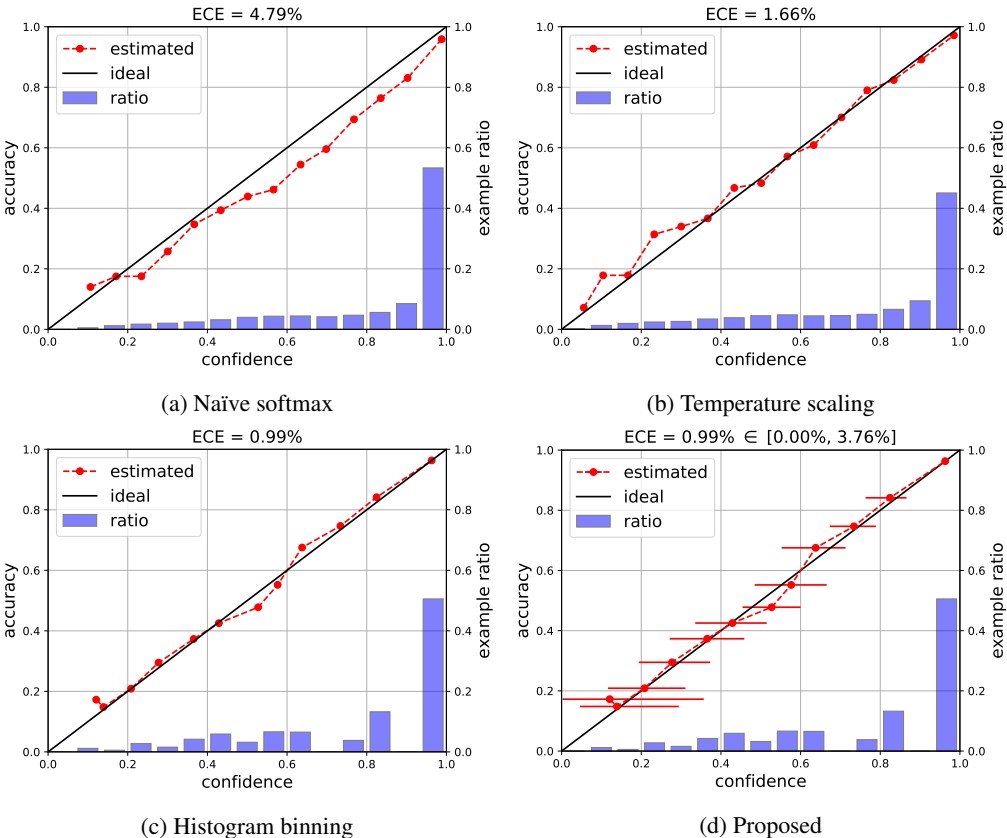

Figure 5: Calibration comparison in reliability diagrams and ECEs. The size of validation set for calibration is $20,000$. The blue histogram represents the example ratio for the corresponding bin. The diagonal line labeled "ideal" is the best reliability diagram, which produces the zero ECE. The estimated reliability diagram is represented "estimated" in dotted red. (a) The naïve softmax output from a neural network is unreliable in ECE. (b, c) The temperature scaling and histogram binning are fairly good calibration approaches, which decreases ECE. (d) The proposed approach generates "induced" intervals (see Appendix F.3) on top of the histogram binning approach, where each interval contains the ideal diagonal line with high probability. Moreover, the proposed one also produces induced ECE interval, where it contains the zero ECE with high probability.

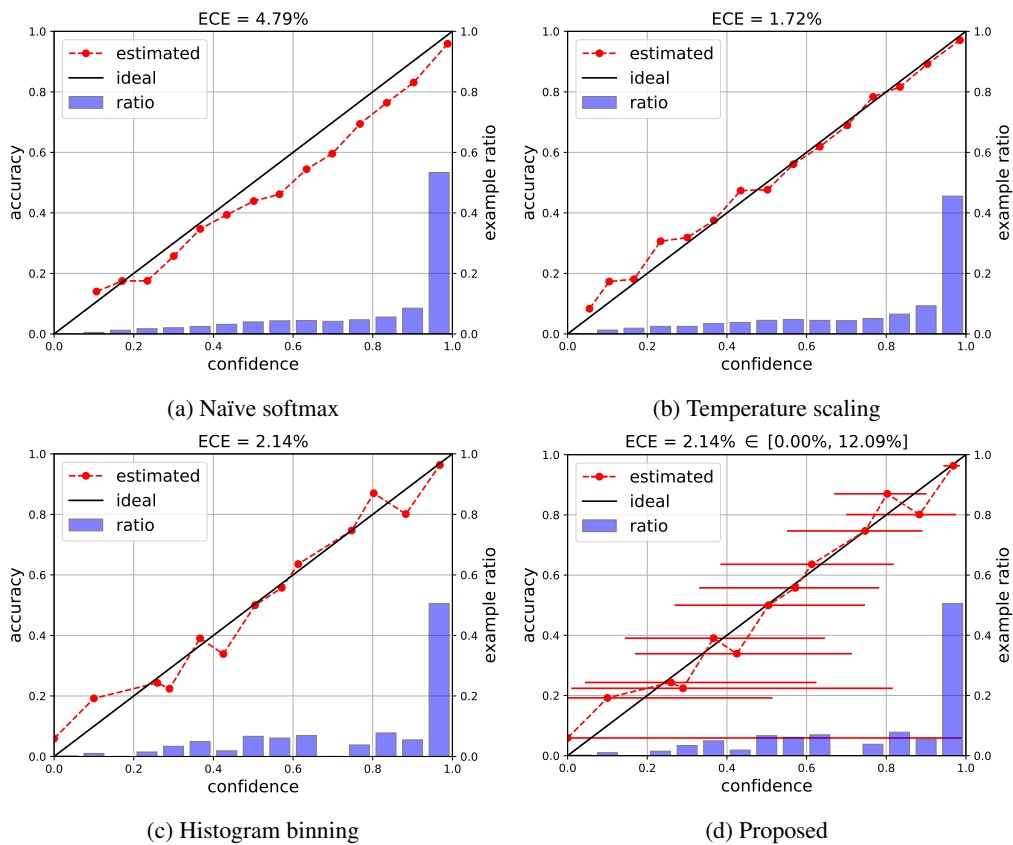

(a) Naïve softmax

(b) Temperature scaling

(c) Histogram binning

(d) Proposed

Figure 6: Calibration comparison via reliability diagrams and ECE. The size of validation set for calibration is $2,000$. See the caption of Figure 5 for interpretation. For the induced intervals, the length of the interval is larger than that with $n = 20,000$ due to the estimation error.

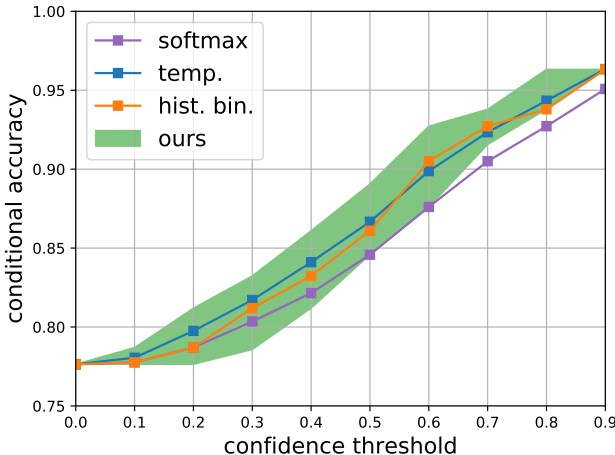

Figure 7: Accuracy-confidence plot. This plot is useful for choosing a proper confidence threshold that achieves a desired conditional accuracy (Lakshminarayanan et al., 2017); the $x$-axis is the confidence threshold $t$ and the $y$-axis is the empirical value of the conditional accuracy $\mathbb{P}\left[\hat{y}(x) = y \mid \hat{p}(x) \geq t\right]$. Since our approach outputs an interval $[\underline{c}(x), \overline{c}(x)]$, we plot $\mathbb{P}\left[\hat{y}(x) = y \mid \underline{c}(x) \geq t\right]$ and $\mathbb{P}\left[\hat{y}(x) = y \mid \overline{c}(x) \geq t\right]$ for the upper and lower bound of the green area, respectively.

## G.2 Fast DNN Inference

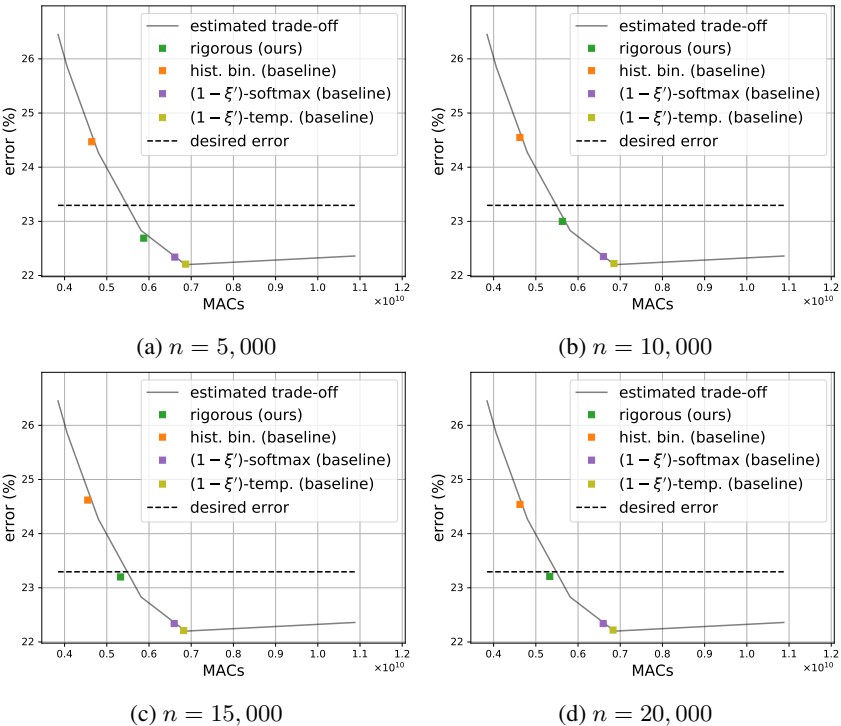

(a) $n = 5,000$

(b) $n = 10,000$

(c) $n = 15,000$

(d) $n = 20,000$

Figure 8: Ablation study on various $n$ for $\xi = 0.02$, $\delta = 10^{-2}$, and $M = 2$. Each plot uses the shown number of samples $n$ with the same $\xi$ and $\delta$. The "estimated trade-off" represents the error and MACs trade-off depending on threshold $\gamma_1$. The markers show the trade-off by the baselines. The "desired error" is a user-specified error bound, where the error of each method need to be below of this line. The proposed approach reduces the inference speed as the number of sample increases, while satisfying the desired error bound. The baselines are either fails to satisfy the desired error bound or overly conservative to satisfy the bound.

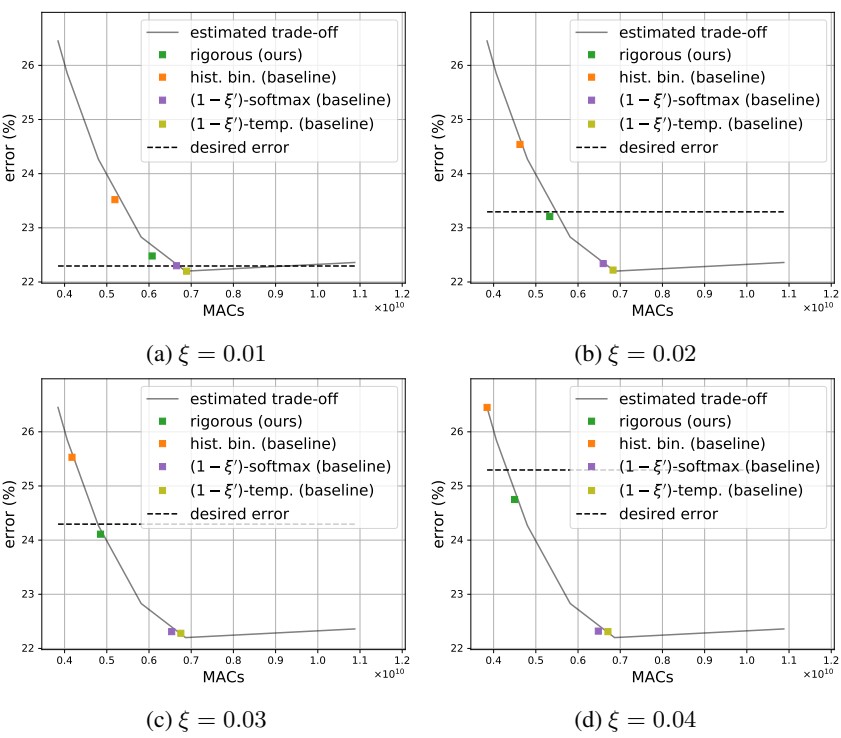

Figure 9: Ablation study on various $\xi$ for $n = 20,000$, $\delta = 10^{-2}$, and $M = 2$. Each plot uses the shown desired relative error $\xi$ with the same $n$ and $\delta$. The "estimated trade-off" represents the error and MACs trade-off depending on threshold $\gamma_1$. The markers show the trade-off by the shown baselines. The "desired error" is a user-specified error bound, where the error of each method need to be below of this line. The proposed approach allows more error as at most specified by the desired error, to reduce the inference speed. Other baselines either overly increase the error or conservatively maintain overly low error.

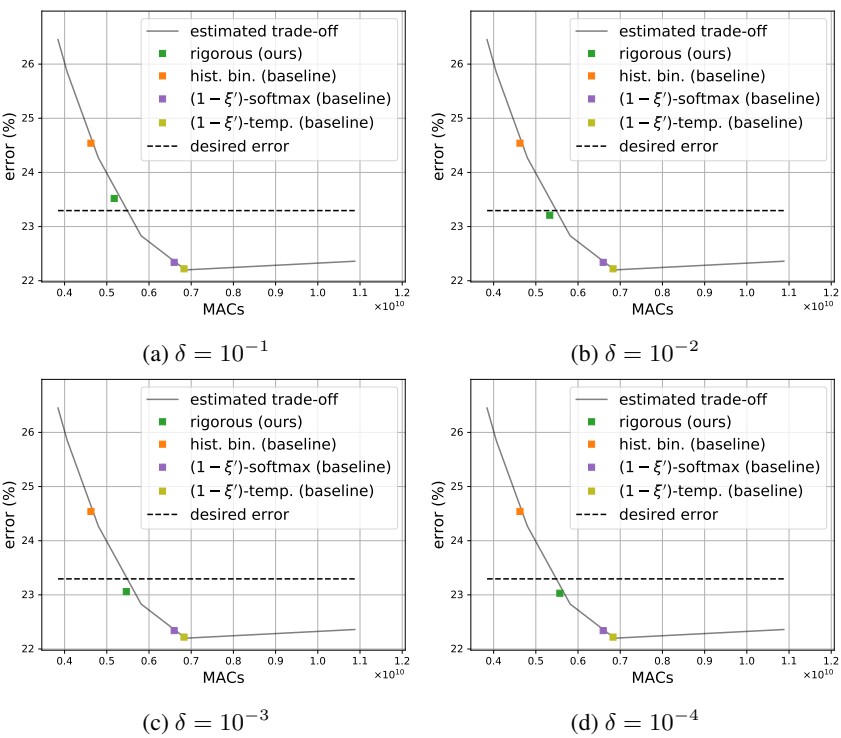

(a) $\delta = 10^{-1}$

(b) $\delta = 10^{-2}$

(c) $\delta = 10^{-3}$

(d) $\delta = 10^{-4}$

Figure 10: Ablation study on various $\delta$ for $n = 20,000$, $\xi = 0.02$, and $M = 2$. Each plot uses the shown misconfidence level $\delta$ with the same $n$ and $\xi$. The "estimated trade-off" represents the error and MACs trade-off depending on threshold $\gamma_1$. The markers show the trade-off by the shown baselines. The "desired error" is a user-specified error bound, where the error of each method need to be below of this line. The proposed approach produces larger error gap toward the desired error as we enforce stronger confidence level—*i.e.,* from $\delta = 10^{-1}$ to $\delta = 10^{-4}$. Note that other baselines do not depend on $\delta$ as designed.

## G.3 SAFE PLANNING

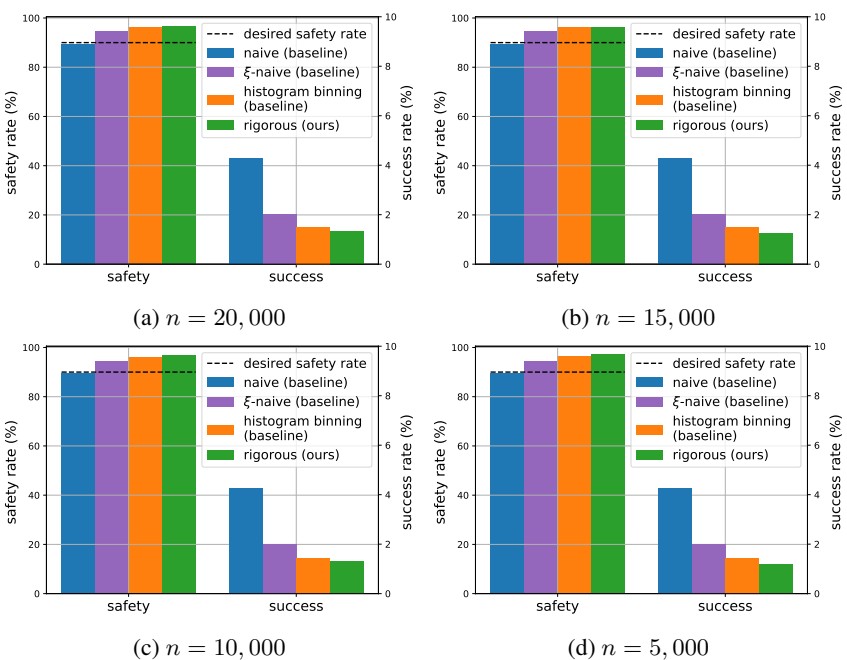

(a) $n = 20,000$

(b) $n = 15,000$

(c) $n = 10,000$

(d) $n = 5,000$

Figure 11: Ablation study on various $n$ for $\xi = 0.1$ and $\delta = 10^{-2}$. Each plot uses the shown sample size $n$ with the same $\xi$ and $\delta$. The "naive", "$\xi$-naive", and "histogram binning" represent baseline results on the safety rate and success rate. The proposed approach is labeled "rigorous". The desired safety rate is a user-specified rate, where the safety rate of each method need to be above of this line. The safety rate of the proposed approach is above the desired safety rate, and it tends to be closer to the desired safety rate as $n$ increases.

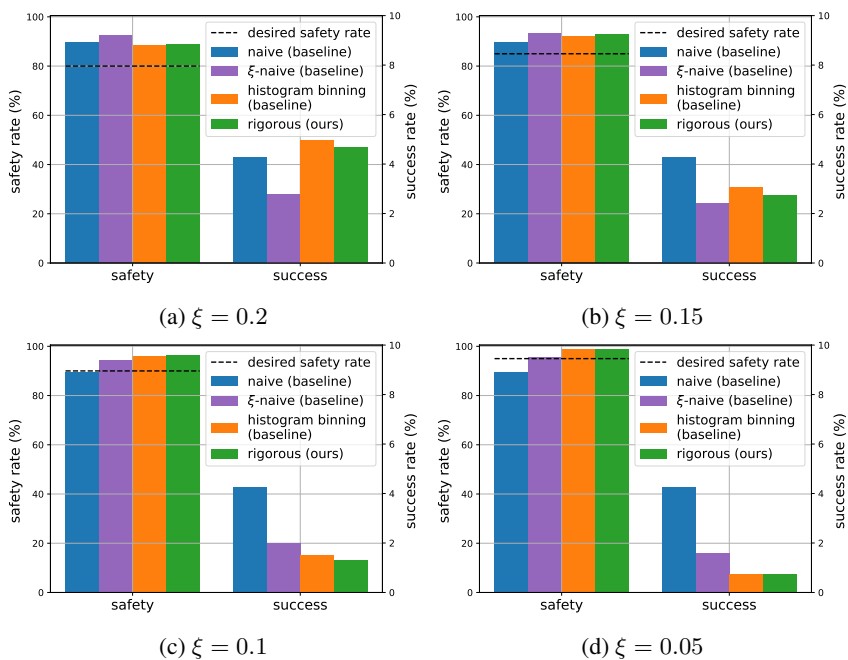

(a) $\xi = 0.2$

(b) $\xi = 0.15$

(c) $\xi = 0.1$

(d) $\xi = 0.05$

Figure 12: Ablation study on various $\xi$ for $n = 20,000$ and $\delta = 10^{-2}$. Each plot uses the shown desired unsafe rate $\xi$ with the same $n$ and $\delta$. The "naive", "$\xi$-naive", and "histogram binning" represent baseline results on the safety rate and success rate. The proposed approach is labeled "rigorous". The desired safety rate is a user-specified rate, where the safety rate of each method need to be above of this line. The safety rate of the proposed approach is closely above the desired safety rate to satisfy the safety constraint. However, the naive can violate the safety constraint, and the $\xi$-naive can be overly optimistic. The histogram binning and $\xi$-naive look empirically fine, but they can in theory violate desired safety rate (*e.g.,* see Figure 4b).

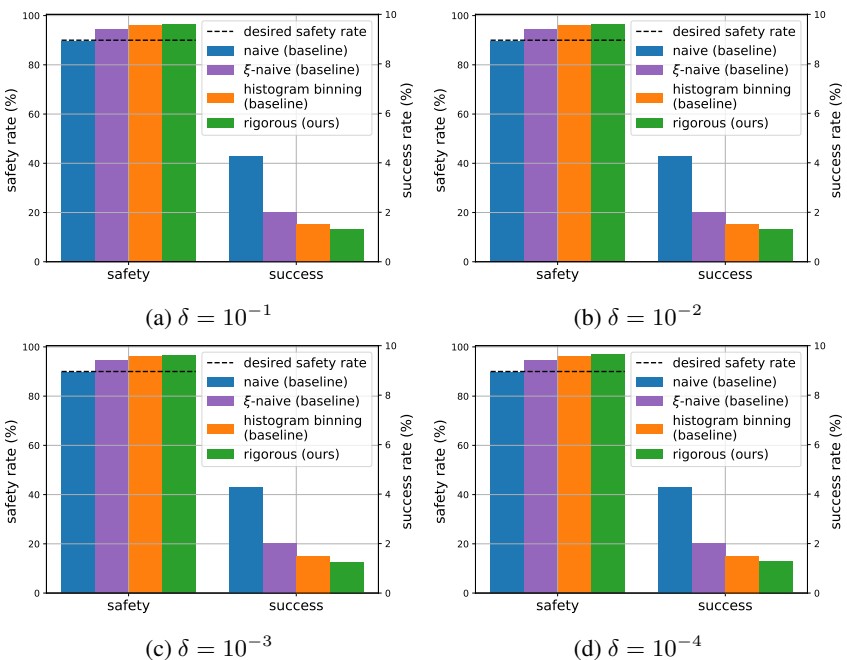

Figure 13: Ablation study on various $\delta$ for $n = 20,000$ and $\xi = 0.1$. Each plot uses the shown misconfidence level $\xi$ with the same $n$ and $\xi$. The "naive", "$\xi$-naive", and "histogram binning" represent baseline results on the safety rate and success rate. The proposed approach is labeled "rigorous". The desired safety rate is a user-specified rate, where the safety rate of each method need to be above of this line. The proposed approach produces more conservative safety rates—*i.e.,* larger gap between the estimated safety rate and desired safety rate—as we enforce stronger confidence level—*i.e.,* from $\delta = 10^{-1}$ to $\delta = 10^{-4}$. Note that other baselines do not depend on $\delta$ as designed.

