# OpenReview forum: "PAC Confidence Predictions for Deep Neural Network Classifiers"
_ICLR.cc/2021/Conference — ICLR 2021 Poster_

### Official Review · AnonReviewer2 · 2020-10-26
**Good idea, good motivation/use cases, needs some clarification + cleaning up**

**Rating:** 6
**Confidence:** 4

**Review:**

Summary: This paper proposes a method for obtaining probably-approximately correct (PAC) predictions given a pre-trained classifier. The PAC intervals are connected to calibration, and take the form of confidence intervals given the bin a prediction falls in. They demonstrate and explore two use cases: applying this technique to get faster inference in deep neural networks, and using the PAC predictor to do safe planning. Experiments in both of these cases show improvements in speed-accuracy or safety-accuracy tradeoffs, as compared to baselines.

I’m recommending acceptance since the idea seems useful and well-argued both conceptually and experimentally. However, the paper needs some work in terms of clarification of the key ideas – with this clarification I can raise my score.

Strong points:
-	The proposed method provides a provable guarantee on the reliability of a pre-trained methods prediction, which is a very nice property to have in the reliability/safety problem.
-	 This approach is a simple but good idea, seems grounded in a good motivation and the explored use cases are informative and interesting.
-	Experimentally, the method shows improvements over a naïve baselines, and demonstrate that it can obey a given error or safety threshold in practice, an important property

Weak points + Clarifications:
-	I am confused about the application of this method to safe planning. In particular, it seems to me like the proposed intervals only hold their PAC guarantee when the test-time distribution matches the training distribution. However, this will not be the case in the safe planning setting as I understand it, since the observed trajectories are drawn from a different policy than the one which will be implemented in the world
-	With respect to these test set questions, it would be nice to see a little discussion in the paper of how these guarantees transfer from training set to the test distribution
-	I would like to see more explanation of the proofs in the appendix, right now they are a little too compact for me
-	Experimental baselines raise some questions for me. First, I need more explanation on the histogram binning baseline beyond the one sentence given. Second, the authors state this baseline “does not satisfy the desired error” – but I’m not sure why we would expect it to, since that baseline was not tuned to any sort of error level. Finally, I would like to see naïve threshold baselines in the safe planning setting for more cautious values than 0.5 – since that is more aligned with the goals of safety.
-	In your PAC definition, we could just always output [0, 1] to satisfy it. Therefore, when framing the goals of your method, you should be a little more clear about exactly what you want from a PAC prediction.
-	A number of notation errors throughout which are important to fix for clarity and neatness, and a decent amount of lack of precision in language throughout which is important to fix for clarity. See Other Feedback
-	Line below Eq 5 confuses me: you say you exit at m if \hat{y}_m correctly classified an example also correctly classified at \hat{y}_M. But how can you know this without doing inference to the last layer?
-	Is there a reason why the greedy approach to Fast DNN inference you take is desirable? Be more clear about why you chose this and if it is optimal somehow or not

Other feedback:
-	- Some precision in language could be improved throughout – for instance “by using the accurate model only if the confidence of the accurate one is underconfident” on p2 doesn’t really make sense
-	-p2: “a naively trained DNN is not reliable” what does reliable mean here?
-	-above eq 2, should this be \kappa_x ?
-	-Defn 1: should this be a nearness constraint rather than equality? We have multiple examples x in a bin each with their own p-hat, and so not all of them can be equal to c(x)
-	-Is there a reason why these intervals should be defined as continuous rather than possibly a disjoint set?
-	-should make it clearer from the start that this is defined for post-hoc classifiers, you’re not learning these intervals directly
-	Eq 4 – the bold theta-hat here is different from the one defined in the line above
-	Need more explanation on “The following expression is equivalent due to the relationship between the Binomial and Beta distributions” p. 4
-	In the definition of C-hat, you’re overloading x on the right side of the given sign. Can you use x’ or something?
-	In the “important case” below Thm 1, can you clarify – this is the mean right?
-	Problem formulation in Fast DNN Inference: should the \hat{y}_i be \hat{y}_m? I feel like i is not scoped here
-	You define d_m below the problem formulation but it isn’t in the formulation itself.
-	Top of p5, you train the network in the “standard way” – this is not clear. Do the gradients at the lower levels flow back through the earlier layers? Or are they stopped and the only gradients are from the final prediction task at the last layer? Either could make sense to me
-	Should define more carefully what the “composed classifier” refers to
-	Clarify how rollouts work – will you observe unsafe states? Sometimes in safe planning you assume you don’t actually observe the unsafe states in training but it looks like you need that
-	Bottom of p6: what is Z’? Not sure what Z is – is it ordered pairs? Why is the second element always 1?
-	P13: why is this an upper bound? It’s hard to parse but it looks equal to the expression in 10 at first glance – it’s the appendix so please explain further. You can also note that the E_t are disjoint by definition
-	Also please add comments about the appendix figures? I have no idea what they are

---

> ### Author Response · Authors · 2020-11-18
> **Response to Comments by Reviewer (1/2)**
>
> We appreciate the reviewer’s value comments. For the “Other feedback”, we will directly update them on the paper.
>
> ---
>
> **Comment 1**: I am confused about the application of this method to safe planning. In particular, it seems to me like the proposed intervals only hold their PAC guarantee when the test-time distribution matches the training distribution. However, this will not be the case in the safe planning setting as I understand it, since the observed trajectories are drawn from a different policy than the one which will be implemented in the world
>
> **Response 1**: The reason that our approach works is we assume that as soon as the robot reaches an unsafe state, it permanently switches to the backup policy. Thus, our recoverability predictor never encounters an out-of-distribution sample. If it were to switch back to the learned policy, then as you point out, our PAC guarantee would no longer be satisfied. We will do our best to improve the clarity of this point in our paper.
>
> ---
> **Comment 2**: With respect to these test set questions, it would be nice to see a little discussion in the paper of how these guarantees transfer from training set to the test distribution
>
> **Response 2**: Our guarantees transfer to the test distribution if it is identical to the training/validation distributions. We believe that extending our results to handling covariate shift is an important direction for future work. See the “Common Response” for a more detailed discussion of future directions.
>
> ---
> **Comment 3**: I would like to see more explanation of the proofs in the appendix, right now they are a little too compact for me
>
> **Response 3**: We will add more detailed explanations to the proofs.
>
> ---
> **Comment 4**: Experimental baselines raise some questions for me. First, I need more explanation on the histogram binning baseline beyond the one sentence given. Second, the authors state this baseline “does not satisfy the desired error” – but I’m not sure why we would expect it to, since that baseline was not tuned to any sort of error level. Finally, I would like to see naïve threshold baselines in the safe planning setting for more cautious values than 0.5 – since that is more aligned with the goals of safety.
>
> **Response 4**:
> *Histogram binning*: This algorithm calibrates the predictions of $\hat{f}$ by sorting the calibration examples $(x, y^*)$ into bins $B_i = [b_i, b_{i+1})$ based on their predicted probability $\hat{p}(x) \in B_i$. Then, for each bin, it computes the empirical probability $p_i = Pr[\hat{y}(x) = y^*]$ (i.e., the empirical counterpart of Eq. 2 in our paper). Finally, it returns predicted probability $p_i$ for all future test instances $x$ such that $\hat{p}(x) \in B_i$.
>
> *Baseline quality*: Histogram binning asymptotically satisfies the desired error bound due to the way it is constructed and the definition of calibration in Eq. 2. However, the estimation error of the bin probability p_i causes it to violate the desired guarantee. In contrast, our approach accounts for the estimation error, enabling us to satisfy the desired guarantee with high probability.
>
> *The threshold of naïve threshold baselines*: We will add results with more cautious and adaptive threshold (i.e., $1 - \epsilon$).
>
> We will clarify all of the aforementioned points in our paper.
>
>
> ---
> **Comment 5**: In your PAC definition, we could just always output [0, 1] to satisfy it. Therefore, when framing the goals of your method, you should be a little more clear about exactly what you want from a PAC prediction.
>
> **Response 5**: Thank you for pointing this out. Our goal is to minimize the size of the interval $\hat{C}$ while satisfying the PAC constraint. We will clarify this point in our problem formulation section.
>
> ---
> **Comment 6**: A number of notation errors throughout which are important to fix for clarity and neatness, and a decent amount of lack of precision in language throughout which is important to fix for clarity. See Other Feedback
>
> **Response 6**: We appreciate the detailed feedback, and will enhance clarity based on the feedback.
>
>
> ---
> **Comment 7**: Line below Eq 5 confuses me: you say you exit at m if \hat{y}_m correctly classified an example also correctly classified at \hat{y}_M. But how can you know this without doing inference to the last layer?
>
> **Response 7**: Sorry for the confusion. Yes, we need to do inference in the last layer to know this fact with certainty. But, if the confidence of $\hat{y}_m$ is sufficiently high (using our confidence intervals to be conservative), then $\hat{y}_m = y^*$ with high probability. In this case, $\hat{y}_M$ either correctly classifies or misclassifies the same example: if the example is misclassified, it does not hurt the relative error, and if it is correctly classified, we have $\hat{y}_m = y^* = \hat{y}_M$ with high probability. We will clarify this point in our paper.

---

> > ### Comment · AnonReviewer2 · 2020-11-18
> > **Response (1/2)**
> >
> > Comment 1: still unclear on this - it seems like the safe planning problem is off-policy planning. Regardless of whether or not you enter an unsafe state, isn't this still a different distribution over trajectories?
> >
> > Comment 2: mentioned this also in the general comment, but I'm interested in understanding how well they transfer to the test set. Does it generally take a lot of data? Do they converge quickly? Not looking for a full proof but some discussion would be pertinent.
> >
> > Comment 3: Okay, I will keep an eye out
> >
> > Comment 4: You say that histogram binning satisfies the desired error bound due to the way it is constructed - what does this mean?

---

> > > ### Author Response · Authors · 2020-11-23
> > > **Response of the additional comments (1/2)**
> > >
> > > Thanks for the swift comments! The following includes the responses of your additional comments. Please feel free to let us know if further clarification is needed.
> > >
> > > ---
> > >
> > > **Comment 1**: still unclear on this - it seems like the safe planning problem is off-policy planning. Regardless of whether or not you enter an unsafe state, isn't this still a different distribution over trajectories?
> > >
> > > **Response 1**: To clarify, we assume that the learned policy is the same at both training time and test time, so it is not exactly an off-policy planning problem. The only thing that changes is that we override the learned policy if we decide it is no longer safe to use. The distribution over states is identical up until (and including) this event. After this event, the robot safely comes to a stop, so it is guaranteed to be safe even though the states visited are now from a different distribution than the training distribution.
> > >
> > > ---
> > >
> > > **Comment 2**: mentioned this also in the general comment, but I'm interested in understanding how well they transfer to the test set. Does it generally take a lot of data? Do they converge quickly? Not looking for a full proof but some discussion would be pertinent.
> > >
> > > **Response 2**: Non-exact intervals (i.e., ones that are only asymptotically correct and do not have finite-sample guarantees) converge at a rate of $1/\sqrt{n}$. While we do not know of analytical bounds on the rate of convergence of CP intervals, we expect that they would converge at the same rate, since asymptotically they should be the same.
> > >
> > > ---
> > >
> > > **Comment 4**: You say that histogram binning satisfies the desired error bound due to the way it is constructed - what does this mean?
> > >
> > > **Response 4**: What we mean is that histogram binning satisfies the desired error bound asymptotically. In particular, if infinitely many samples are collected to construct the histogram binning, then the value in each bin is the same as the true confidence -- i.e., $\hat{p}(x) = c_{\hat{f}}(x)$; thus histogram binning is well-calibrated in the sense of Definition 1. In this case, the histogram binning can satisfy a desired error. However, this property relies on having an infinite number of samples, which does not hold in practice.

---

> > > > ### Comment · AnonReviewer2 · 2020-11-23
> > > > **Policy**
> > > >
> > > > Comment 1: So how are the trajectories obtained? I assume that at training time we can't do any unsafe actions either, so how do we get estimates of their safety? Or at train time are we allowed to go to unsafe states?

---

> > > > > ### Author Response · Authors · 2020-11-23
> > > > > **Response on Policy**
> > > > >
> > > > > **Response to Comment1**: Yes, at training time, the agent is allowed to go to unsafe states. In particular, when collecting calibration sets (i.e., $Z'$ and $W'$), we collect a rollout that includes the first unsafe state. We believe this is a reasonable strategy -- in practice, RL is usually used to train a policy in simulation that is to be deployed on a real robot. Our goal is to ensure that the policy is at least safe with respect to the model -- for instance, currently, we cannot even guarantee that a trained policy does not run into a wall in the simulator. We will update our paper to clarify that we are not addressing the sim-to-real gap (except to the extent that it can be modeled as noise in the simulator).

---

> ### Author Response · Authors · 2020-11-18
> **Response to Comments by Reviewer (2/2)**
>
>
>
> **Comment 8**: Is there a reason why the greedy approach to Fast DNN inference you take is desirable? Be more clear about why you chose this and if it is optimal somehow or not
>
> **Response 8**: Based on your suggestion, we have proved that the current greedy approach is in fact optimal. Intuitively, we are always better off (in terms of inference time) by classifying more inputs using the faster model. We will add a discussion to our paper and a proof of optimality to the appendix.
>
>
> ---
> **Comment 9**: Defn 1: should this be a nearness constraint rather than equality? We have multiple examples x in a bin each with their own p-hat, and so not all of them can be equal to c(x)
>
> **Response 9**: We note that $c_{\hat{f}}$ in Definition 1 is *different* from $c_{\hat{f}}$ in Eq. 1. Instead, we are re-defining $c_{\hat{f}}$ to be the average probability for $x$’s in bin $B_k$, where $k = \kappa_{\hat{f}(x)}$. Thus, by definition, $c_{\hat{f}}(x)$ is the same for all $x$ in this bin. We apologize for the confusion, and will clarify this point in our paper.

---

> > ### Comment · AnonReviewer2 · 2020-11-18
> > **Response (2/2)**
> >
> > Comment 8: I'll keep an eye out for the updated appendix with the proof!

---

### Official Review · AnonReviewer1 · 2020-10-28
**Certainly interesting, but perhaps needs to address limitations more clearly**

**Rating:** 7
**Confidence:** 2

**Review:**

In this work, the authors provide a method for a posteriori calibration of DNN uncertainty with emphasis on constructing a classifier that has PAC uncertainty guarantees. The authors define a ‘calibrated’ probability prediction to be one such that given (for example) an image labeled as a cat, that the probability assigned to the class label ‘cat’ by the predictor is equivalent to the probability that the classifier correctly predicts images from the class ‘cat.’ The authors seek to create a ‘provably’ correct classifier under iid data assumptions.

Accounting for and re-calibrating the poor uncertainty of neural networks is a central problem for deep learning especially when operating in safety-critical domains and so the work is certainly attempting to attack a worthwhile problem.

One concern I have about the theory laid out by the authors is that they place no conditions on \hat{f}. That is, one could have a classifier which has arbitrarily bad uncertainty calibration. Take for example a predictor \hat{f} which assigns \hat{p}(x) = 0.9 to every input ‘x’ regardless of its ultimate accuracy on the class, then given a new input with unknown label, it is not clear to me exactly how the framework would use the intervals to improve the uncertainty of this classifier, especially given that the class of this new point is unknown. I would appreciate if the authors could help me understand this case as I think it is indicative of a misunderstanding on my part.

Another concern is the fact that getting well calibrated Clopper-Pearson intervals with good statistical guarantees takes a non-trivial number of samples and it appears this would scale with the number of classes. Thus, for a task like ImageNet, the authors inference model would require the set Z to be quite large (quoted at 20000). It seems that considering this overhead would greatly slow the average inference time. Is this overhead considered in figure 2a? I think this is a OK trade-off to make when in safety-critical scenarios, but then the authors give “fast inference” as one of their primary applications, it seems like a bit more discussion of this may be warranted.

The authors use the word “provably” correct in a couple places (page 2 and 3) where the correct term they should use is “probably” or PAC. Saying that a statistical guarantee is “provably” correct is an over claim and in my view this should be corrected.

My last concern is not so much with what the authors have presented, just in the fact that the limitation is not clearly stated. When using statistical guarantees such as those given under the PAC framework, the iid assumption is almost always necessary. Yet, it can be limiting in both the case of planning and image classification, especially when we are considering safety-critical applications. In such applications, one often considers worst-case scenarios and in these cases the iid assumption usually does not hold. Thus, the PAC guarantees in this paper would seemingly be invalidated. For example, adversarial examples may be out of the support of the data distribution, but are still valid inputs. In the presence of adversarial examples the uncertainty guarantees presented here are rendered void. Similarly, there is often concept drift in RL and control problems. Ultimately, I don’t think this is only a minor mark against the work, and one that can be overlooked given that such cases are at least clearly stated as limitations of the approach.


Post-Rebuttal Comment:

I would like to thank the authors for thoughtfully answering my concerns and questions. I think the small modifications made in response to my comments have made the paper much easier to understand and I think the work is well presented and positioned. Ultimately, I have increased my score to Accept on the basis that I no longer have any major criticism of the work. I do hope the authors can make a more prominent note about appendix F1 in the main text as I think it is an interesting and important thing to highlight for those who may be skeptical.

---

> ### Author Response · Authors · 2020-11-18
> **Response to Comments by Reviewer**
>
> Thanks for the valuable discussion. The following are the answers to your concerns. Let us know if further clarification is needed.
>
> ---
>
> **Comment 1**: One concern I have about the theory laid out by the authors is that they place no conditions on $\hat{f}$. That is, one could have a classifier which has arbitrarily bad uncertainty calibration. Take for example a predictor $\hat{f}$ which assigns $\hat{p}(x) = 0.9$ to every input $x$ regardless of its ultimate accuracy on the class, then given a new input with unknown label, it is not clear to me exactly how the framework would use the intervals to improve the uncertainty of this classifier, especially given that the class of this new point is unknown.
>
> **Response 1**: In fact, ours can be applied to any $\hat{f}$. If $\hat{f}$ has poor calibration, then our algorithm will simply rescale the probabilities to be very uncertain (as is the case with the histogram binning algorithm underlying our approach). Intuitively, the interval we produce captures the uncertainty in the calibrated probabilities. In the reviewer’s example, all the examples will fall in a single bin B. Then, histogram binning will rescale the confidence of this bin is the empirical accuracy of examples in this bin -- e.g., around $\hat{p}(x) = 0.5$ if $\hat{f}$ is random. Our contribution is to additionally provide a confidence interval around this value. We will clarify this point in our paper.
>
> ---
>
> **Comment 2**: ...Thus, for a task like ImageNet, the author's inference model would require the set Z to be quite large (quoted at 20000). It seems that considering this overhead would greatly slow the average inference time. Is this overhead considered in figure 2a?
>
> **Response 2**: Our approach constructs the intervals for each bin during design time. They do not need to be re-computed for new test instances. In particular, the overhead is negligible and O(1) (i.e., simply a table lookup for the bin containing the predicted probability). This overhead is considered in Figure 2c when measuring CPU and GPU times. We will clarify this point in our paper.
>
> ---
>
> **Comment 3**: The authors use the word “provably” correct in a couple places (page 2 and 3) where the correct term they should use is “probably” or PAC. Saying that a statistical guarantee is “provably” correct is an over claim and in my view this should be corrected.
>
> **Response 3**: We meant “provably correct” in the sense of “provable PAC guarantee”. To avoid confusion, we will change the term “provably” to “PAC” throughout our paper.
>
> ---
>
>
> **Comment 4**: My last concern is not so much with what the authors have presented, just in the fact that the limitation is not clearly stated. When using statistical guarantees such as those given under the PAC framework, the iid assumption is almost always necessary. Yet, it can be limiting in both the case of planning and image classification, especially when we are considering safety-critical applications. In such applications, one often considers worst-case scenarios and in these cases the iid assumption usually does not hold. Thus, the PAC guarantees in this paper would seemingly be invalidated. For example, adversarial examples may be out of the support of the data distribution, but are still valid inputs. In the presence of adversarial examples the uncertainty guarantees presented here are rendered void. Similarly, there is often concept drift in RL and control problems. Ultimately, I don’t think this is only a minor mark against the work, and one that can be overlooked given that such cases are at least clearly stated as limitations of the approach.
>
> **Response 4**: We will explicitly state the assumption in our introduction and describe its limitations in terms of applications to reinforcement learning, concept drift, and adversarial examples. We agree that providing theoretical guarantees for out-of-distribution data is an important direction; however, we believe that our work is an important stepping stone towards this goal. See the “Common Response” for a more detailed discussion of future directions.

---

### Official Review · AnonReviewer4 · 2020-11-05
**Method to provide prediction guarantees of DNNs, but some crucial experimental results are missing.**

**Rating:** 6
**Confidence:** 4

**Review:**

This is a paper that focusses on the timely and important problem of uncertainty quantification for the predictions of deep neural network classifiers. The authors propose constructing calibrated outputs that have provable correctness guarantees, using PAC-style arguments.

The authors also demonstrate how this framework can be utilized for computationally efficient predictions by combining a smaller, faster albeit less accurate model with a larger, more accurate model, utilizing the latter only when the former is less confident. For this to work, one needs good guarantees on the DNN's estimates of its confidence -- and creating such guarantees forms the crux of the paper.

Pros:
+ Paper is well written
+ Important and timely problem, motivating arguments are well constructed
+ Paper appears to be mathematically sound though I did not check all the proofs in the appendix.

Cons
+ One of the crucial assumptions is that the data during test time will be from in-distribution. While I understand it is hard and maybe impossible to provide any guarantees for out-of-distribution data,  it is important to realize that one of the most common ways in which DNNs can fail when deployed in safety critical systems is when faced with predicting from an out-of-distribution data. So it is unclear how  practically applicable the proposed method is.

+ If somehow, one is always guaranteed to work within in-distribution data, then the authors should compare other methods that improve calibration (but don't have guarantees) with the proposed method, both in terms of calibration and computational efficiency in the fast-model/slow-model approach.

+ Also I do not see any calibration performance in the experimental results. While I understand the proposed method is using histogram binning and is not a new calibration method per se, these results should be included. That is,show how  accuracy correlates with the softmax predictions (on the test set) using proposed technique.

+ Since the proposed method provides stronger UQ than other methods, I would also like to see accuracy-vs-coverage curves for benchmark datasets, and compare this to such curves for the baseline (where one thresholds on the winning softmax scores).

Overall, this is a very worthwhile line of work, and I feel the paper has merit, but given that some important results are missing, as it stands, does not meet the bar for acceptance.

== Post rebuttal update ==

See my response to author's rebuttal below. In light of new experimental results, I feel this now meets the bar of acceptance at ICLR, and hence  updating my score to 6.

---

> ### Author Response · Authors · 2020-11-18
> **Response to Comments by Reviewer**
>
> Thank you for the valuable comments. We have done our best to address them below; please let us know further clarification is needed.
>
> ---
>
> **Comment 1**: One of the crucial assumptions is that the data during test time will be from in-distribution. While I understand it is hard and maybe impossible to provide any guarantees for out-of-distribution data, it is important to realize that one of the most common ways in which DNNs can fail when deployed in safety critical systems is when faced with predicting from an out-of-distribution data. So it is unclear how practically applicable the proposed method is.
>
> **Response 1**:We also believe that providing theoretical guarantees for out-of-distribution data is an important direction; however, we believe that our work is an important stepping stone towards this goal. See the “Common Response” for a more detailed discussion of future directions.
>
> ---
>
> **Comment 2**: If somehow, one is always guaranteed to work within in-distribution data, then the authors should compare other methods that improve calibration (but don't have guarantees) with the proposed method, both in terms of calibration and computational efficiency in the fast-model/slow-model approach.
>
> **Response 2**: We note that we have already compared with histogram binning in our experiments. For the computational efficiency experiments (Figure 2), our proposed approach helps to satisfy the downstream guarantee whereas histogram binning does not. We will add comparisons to temperature scaling as well; we are also happy to add other approaches as suggested by the reviewer.
>
> ---
>
> **Comment 3**: Also I do not see any calibration performance in the experimental results. While I understand the proposed method is using histogram binning and is not a new calibration method per se, these results should be included. That is,show how accuracy correlates with the softmax predictions (on the test set) using proposed technique.
>
> **Response 3**: Our approach does not provide calibrated predictions that are directly comparable to histogram binning, since it is providing lower and upper bounds on the bin probabilities. Nevertheless, we will add comparisons in terms of reliability diagrams for histogram binning where we include our confidence intervals around the predicted probability for that bin, as well as comparisons to uncalibrated probabilities and temperature scaling. This comparison allows us to check whether the proposed approach is good in terms of ECE---i.e., the induced intervals pass the diagonal of the reliability diagram most of the time. We note that our confidence intervals need to be remapped to make sense in the reliability diagram, so we refer to the intervals shown as “induced intervals”. We will add a detailed description in the paper.
>
> ---
>
> **Comment 4**: Since the proposed method provides stronger UQ than other methods, I would also like to see accuracy-vs-coverage curves for benchmark datasets, and compare this to such curves for the baseline (where one thresholds on the winning softmax scores).
>
> **Response 4**: Could you clarify the meaning of “accuracy-vs-coverage curves for benchmark datasets”? It would be also great if we could have an example of a paper that has the related experiments.

---

> ### Author Response · Authors · 2020-11-25
> **Update for Response to Comments by Reviewer**
>
> ***Comment 4***: Since the proposed method provides stronger UQ than other methods, I would also like to see accuracy-vs-coverage curves for benchmark datasets, and compare this to such curves for the baseline (where one thresholds on the winning softmax scores).
>
> ***Response 4 (updated)***: We have added an accuracy-coverage plot for our approach along with baselines in Figure 7 in Appendix G.

---

### Author Response · Authors · 2020-11-18
**Common Response to Comments by all reviewers**

Dear reviewers,

We appreciate your valuable feedback. The following includes changes we are making, and discussion on limitations and possible extensions of our approach to handle out-of-distribution test examples.


**Changing list**
- (clarity) We will enhance overall clarity of the paper
- (experiment) We will add evaluation and comparison results in calibration performance
- (experiment) We will add additional comparison results in fast DNN inference
- (experiment) We will add additional comparison results in safe planning
- (proof) We will add an optimality proof of the greedy algorithm in fast DNN inference, along with the corresponding discussion
- (discussion) We will add additional discussions mentioned in the response---We are still working on updating the paper to address the reviewer comments, and will update our paper as soon as possible.



**Discussion on out-of-distribution cases**

*Related comments:*
- *Comment 1*: One of the crucial assumptions is that the data during test time will be from in-distribution. While I understand it is hard and maybe impossible to provide any guarantees for out-of-distribution data, it is important to realize that one of the most common ways in which DNNs can fail when deployed in safety critical systems is when faced with predicting from an out-of-distribution data. So it is unclear how practically applicable the proposed method is.
- *Comment 2*: My last concern is not so much with what the authors have presented, just in the fact that the limitation is not clearly stated. When using statistical guarantees such as those given under the PAC framework, the iid assumption is almost always necessary. Yet, it can be limiting in both the case of planning and image classification, especially when we are considering safety-critical applications. In such applications, one often considers worst-case scenarios and in these cases the iid assumption usually does not hold. Thus, the PAC guarantees in this paper would seemingly be invalidated. For example, adversarial examples may be out of the support of the data distribution, but are still valid inputs. In the presence of adversarial examples the uncertainty guarantees presented here are rendered void. Similarly, there is often concept drift in RL and control problems. Ultimately, I don’t think this is only a minor mark against the work, and one that can be overlooked given that such cases are at least clearly stated as limitations of the approach.
- *Comment 3*: With respect to these test set questions, it would be nice to see a little discussion in the paper of how these guarantees transfer from training set to the test distribution

*Common response*:

We also believe that providing theoretical guarantees for out-of-distribution data is an important direction; however, we believe that our work is an important stepping stone towards this goal. In particular, to the best of our knowledge, we do not know of any existing work that provides theoretical guarantees on calibrated probabilities even for the in-distribution case.

One possible direction is to use our approach in conjunction with covariate shift detectors (e.g., [1]). Alternatively, it may be possible to directly incorporate ideas from recent work on calibrated prediction with covariate shift [2] or uncertainty set prediction with covariate shift [3, 4]. In particular, we can use importance weighting q(x)/p(x), where p(x) is the training distribution and q(x) is the test distribution, to reweight our training examples, enabling us to transfer our guarantees from the training set to the test distribution. The key challenge is when these weights are unknown. In this case, we can estimate them given a set of unlabeled examples from the test distribution [2], but we then need to account for the error in our estimates. We will add discussion on this future direction to our paper.

[1] A. Gretton, K. M. Borgwardt, M. J. Rasch, B. Schölkopf, A. Smola. A kernel two-sample test. JMLR 2012.

[2] S. Park, O. Bastani, J. Weimer, and I. Lee. Calibrated Predictions with Covariate Shift via Unsupervised Domain Adaptation. AISTATS 2020

[3] M. Cauchois, S. Gupta and A. Ali and J. C. Duchi. Robust Validation: Confident Predictions Even When Distributions Shift. 2020

[4] R. F. Barber, E. J. Candes, A. Ramdas, R. J. Tibshirani. Conformal prediction under covariate shift. NeurIPS 2019.

---

> ### Comment · AnonReviewer2 · 2020-11-18
> **Comment 3 clarification**
>
> I wrote comment 3 here - I wasn't actually asking about test OOD data but rather test data drawn from the same distribution. Since the intervals are derived from properties of the training distribution, how well will they work on an example drawn at test time? I understand this is potentially a can of worms (ie I don't need a fully fledged generalization bound) but I'm curious for some discussion of this topic

---

> > ### Author Response · Authors · 2020-11-25
> > **Response to Comment 3 clarification**
> >
> > Yes, our bound is a generalization bound -- it says that the confidence intervals are valid for new examples drawn from the test distribution (assuming it equals the training distribution). In particular, it says that with probability at least $1 - \delta$, we construct confidence intervals that are valid for all x.

---

### Author Response · Authors · 2020-11-25
**Update for the Common Response to Comments by all reviewers**

Dear reviewers,

We have updated our paper based on your feedback. List of changes:

* (clarity) We did our best to enhance overall clarity of the paper
* (experiment) We added evaluation and comparison results on calibration performance (Section 5.1 and Appendix G.1)
* (experiment) We added additional baselines in fast DNN inference (e.g., Figure 3)
* (experiment) We added additional baselines in safe planning (e.g., Figure 4)
* (proof) We added an optimality proof of the greedy algorithm in fast DNN inference for the case M=2 in Theorem 3
* (discussion) We added additional discussions mentioned in the response (across paper and in Appendix F)

Please let us know if you have any additional concerns.

---

### Decision · Program_Chairs · 2021-01-07
**Final Decision**

**Decision:**

Accept (Poster)

**Comment:**

The paper provides a method for constructing PAC confidence scores for pre-trained deep learning classifiers. The reviewers were all positive about the paper.

Pros:
- Has provable guarantees on the reliability of the prediction. Such guarantees are quite desirable in practice.
- The problem of neural network uncertainty is important and timely problem, especially in safety-critical applications.
- The method is simple and well-motivated.
- Strong empirical performance.
- Interesting applications to fast DNN inference and safe planning.

Cons:
- Lack of generalization guarantees-- the guarantees in the paper only hold on the training set; but in practice, performance in test is what's important.
- Only a handful of baselines tested against, most of which (if not all) were naive.